# RG/RGG repeats in the *C. elegans* homologs of Nucleolin and GAR1 contribute to sub-nucleolar phase separation

Emily L. Spaulding [1] ✉, Alexis M. Feidler[1], Lio A. Cook[1] & Dustin L. Updike [1]

The intrinsically disordered RG/RGG repeat domain is found in several nucleolar and P-granule proteins, but how it influences their phase separation into biomolecular condensates is unclear. We survey all RG/RGG repeats in *C. elegans* and uncover nucleolar and P-granule-specific RG/RGG motifs. An uncharacterized protein, K07H8.10, contains the longest nucleolar-like RG/RGG domain in *C. elegans*. Domain and sequence similarity, as well as nucleolar localization, reveals K07H8.10 (NUCL-1) to be the homolog of Nucleolin, a protein conserved across animals, plants, and fungi, but previously thought to be absent in nematodes. Deleting the RG/RGG repeats within endogenous NUCL-1 and a second nucleolar protein, GARR-1 (GAR1), demonstrates these domains are dispensable for nucleolar accumulation. Instead, their RG/RGG repeats contribute to the phase separation of proteins into nucleolar sub-compartments. Despite this common RG/RGG repeat function, only removal of the GARR-1 RG/RGG domain affects worm fertility and development, decoupling precise sub-nucleolar structure from nucleolar function.

The concentration of molecules into biomolecular condensates (BMCs) through the process of liquid-liquid phase separation (LLPS) is a strategy used by cells to promote control over biological reactions[1]. Some BMCs, such as nucleoli and *C. elegans* germ granules (P-granules), are thought to be multi-layered condensates, comprised of distinct and co-existing liquid phases[2–8]. The multi-condensate structure may facilitate ribosome biogenesis in nucleoli and post-transcriptional regulation in P-granules, but little in vivo work has been done to test this hypothesis[9–13].

How LLPS of individual proteins is controlled to result in the complex assembly and internal organization of BMCs is unclear. Many core BMC proteins contain intrinsically disordered domains that promote LLPS, including a variety consisting of arginine-glycine (RG/RGG) repeats[14–16]. Humans have over 1800 proteins with at least two closely spaced RG or RGG motifs, several of which are core components of nucleoli and germ granules[17]. For example, the RG/RGG repeat-containing protein, Nucleolin (NCL), is one of the most abundant nucleolar proteins. Within nucleoli NCL functions in rRNA transcription and processing[18–21]. Fibrillarin (FBL) and GAR1 are two additional nucleolar RG/RGG proteins that are highly conserved from yeast to humans[22,23]. Both proteins are components of small nucleolar ribonucleoproteins (snoRNPs) that function in rRNA modification. FBL is a member of the C/D box snoRNPs that direct 2′-O-methylation of rRNA, while GAR1 is a member of the H/ACA snoRNP family that is responsible for 18 S rRNA production and rRNA pseudouridylation[24,25].

The function of RG/RGG repeats in LLPS of proteins into BMCs, especially in vivo, is unclear, and their role in the formation of sub-condensates is largely unexplored. For example, the FBL RG/RGG domain is not required for nucleolar accumulation, and in *Xenopus* oocytes it also appears dispensable for sub-nucleolar localization[3,26]. RG/RGG repeats in GAR1 are also dispensable for nucleolar accumulation[27]. In contrast, RG/RGG repeats in *Xenopus* and mouse NCL are required for complete nucleolar accumulation, although some Ncl still accumulates in mouse neuronal nucleoli when the RG/RGG domain is absent[28,29]. Increasing evidence links disrupted LLPS to human disease, including cancer and some of the most severe forms of neurodegeneration[30]. Uncovering the mechanisms of LLPS in vivo will provide a greater understanding of its dysregulation in disease.

As an optically clear organism with efficient CRISPR/Cas9 genetic editing capabilities, *C. elegans* provide an opportunity to study how

---

[1]Davis Center for Regenerative Biology and Medicine, The Mount Desert Island Biological Laboratory, Bar Harbor, ME, USA. ✉e-mail: espaulding@mdibl.org

RG/RGG repeats contribute to LLPS in a living animal. In this study, we identified all RG/RGG repeat-containing proteins in *C. elegans* and discovered the NCL and GAR1 homologs (NUCL-1 and GARR-1). Deleting the RG/RGG repeats from endogenous NUCL-1 and GARR-1 revealed these domains are not required for nucleolar accumulation, but rather for phase separation into sub-nucleolar condensates. Deletion of nucleolar RG/RGG repeats produces viable worms with distinct fertility and developmental phenotypes, providing valuable in vivo models for the continued study of RG/RGG repeat function.

## Results

### RG/RGG repeat survey reveals nucleolar and P-granule sequence motifs

To identify RG/RGG repeats in *C. elegans*, we mined the proteome for three or more occurrences of RG followed by any number of additional G residues and separated by no more than 15 amino acids (Fig. 1a). The 50 longest repeats are found in 43 different proteins and range from 321 to 21 amino acids in length (Supplementary Fig. 1). Repeats are most often located at the C-terminus of proteins but are also found in the middle and N-terminus (Supplementary Fig. 1). SMART protein domain analysis of the 43 proteins revealed co-enrichment of the RNA recognition motif (RRM) (Fig. 1b). Gene ontology analysis revealed that many RG/RGG proteins localize to BMCs, including the nucleolus and P-granules (Fig. 1c). Nucleolar RG/RGG proteins LPD-6, FIB-1 (Fibrillarin), Y66H1A.4 (predicted GAR1 homolog), and RHA-1 all have human homologs with RG/RGG repeats. In contrast, only 1 P-granule protein, LAF-1, has an RG/RGG domain-containing human homolog (Fig. 1d).

To determine if proteins that localize to the same type of BMC contain shared RG/RGG motifs, we separately loaded nucleolar and P-granule RG/RGG sequences into the MEME Motif discovery tool[31]. A phenylalanine-rich motif, "FRGGDRGGFR," was found 2–6 times in each of the RG/RGG sequences from nucleolar proteins FIB-1, LPD-6, and Y66H1A.4 (Fig. 1e)[23]. A similar phenylalanine-rich motif is also present within RG/RGG repeats from the human nucleolar proteins, NCL, FBL, and GAR1 (Fig. 1f). In contrast, a tyrosine-rich motif, "RGGRGGYRGGD," is common to RG/RGG sequences from the P-granule proteins LAF-1, PGL-1, PGL-3, CSR-1, and SNR-3 (Fig. 1g). Therefore, both nucleolar and P-granule RG/RGG repeats contain distinctive motifs, suggesting they may be important for recruitment or organization of proteins into specific BMCs.

### RG/RGG repeat protein K07H8.10 shows homology to human NCL

Although several previously known RG/RGG-containing proteins were identified in our search, many of the longest RG/RGG repeats are found in uncharacterized proteins. For example, the highly abundant K07H8.10 transcript encodes the second-longest RG/RGG domain in *C. elegans*, with 32 RG/RGG repeats within 176 amino acids of sequence[32]. In addition to its N-terminal RG/RGG domain, SMART domain analysis identified a coiled-coil acidic domain and two C-terminal RRMs. The human nucleolar protein, NCL, contains these same 3 domains, although rearranged (Fig. 1h). This rearrangement is only observed in ecdysozoans, including nematodes, tardigrades, and some arthropods (Supplementary Fig. 2). A default BLAST search fails to recognize homology between K07H8.10 and NCL because of the domain reordering. However, the HHpred homology detection tool predicts K07H8.10 as the NCL homolog with 99.97% probability[33]. The MARRVEL toolkit also predicts this homology[34]. Pairwise sequence alignment of individual K07H8.10 and NCL domains using the EMBOSS Water tool reveals 43.3% identity and similarity between the RG/RGG domains, 56.5% identity and 91.3% similarity between the acidic domain, and 28.1% identity and 59.4% similarity between RRMs (Fig. 1h)[35]. In addition, the K07H8.10 RG/RGG repeat contains the phenylalanine-rich nucleolar motif 9 times, predicting association with the nucleolus (Fig. 1e).

To test if K07H8.10 localizes to the nucleolus in living worms, we utilized a split-GFP approach to selectively visualize the endogenous protein in the germline for ease of downstream analysis[36]. The adult germline contains a well-defined progression of mitotic stem cells to meiotic gametes that is easily observable in living worms. CRISPR/Cas9 genome editing was used to place a small, 16 amino acid split superfolder-GFP11 tag (sGFP11) on the C terminus of endogenous K07H8.10 (K07H8.10::sGFP11) in a strain carrying GLH-1::T2A::sGFP(1-10) to drive sGFP(1-10) in the germline. Upon T2A self-cleavage, sGFP(1-10) is released from GLH-1 and associates with K07H8.10-tethered GFP11, resulting in K07H8.10::sGFP1-11 expression in the germline (Fig. 2a). This tagging approach revealed that K07H8.10 accumulates in all germ cell nucleoli at every stage of worm development (Fig. 2b). Although the majority of NCL localizes to the nucleolus, it can be found in the cytoplasm or at the cell surface[37,38]. Confocal DIC imaging shows K07H8.10 is exclusively found in the nucleolus of pachytene germ cells and oocytes (Fig. 2c). Nucleolar vacuoles, which are thought to indicate high nucleolar activity, are observed in some oocyte and pachytene nucleoli[39].

Based on the domain and sequence similarities between NCL and K07H8.10, as well as its nucleolar localization, we renamed K07H8.10 NUCL-1 (NUCLeolin homolog). NCL genes are found in organisms ranging from yeast to plants and mammals and NCL is tightly linked to human disease[40]. NCL has a growing association with the most common familial form of ALS, caused by hexanucleotide repeat expansions in the *C9orf72* gene. For example, NCL binds RNA G-quadruplexes formed by repeat expansion transcripts, colocalizes with these RNA repeats in the motor cortex of ALS patients, and is mis-localized in patient lymphocytes and iPSC-derived neurons[41]. In addition, NCL overexpression is a common feature of tumor cells and cell surface NCL has emerged as an anti-cancer therapy[42–44]. Identification of the *C. elegans* NCL homolog provides a new in vivo model for studying its function in cell biology and its roles in human neurodegenerative disease and cancer[41,44–47].

### Sub-nucleolar compartmentalization of RG/RGG proteins

Nucleolar sub-structure in nematodes is largely unstudied, and the number and composition of sub-compartments has been unclear. To determine if NUCL-1 displays sub-nucleolar compartmentalization, we performed super-resolution confocal imaging of living adult worms. Free GFP shows no sub-nucleolar patterning or organization (Supplementary Fig. 3). In contrast, tagged NUCL-1 displays striking organization in oocyte and pachytene germ cell nucleoli (Fig. 2d). To create a more complete map of nucleolar substructure we fluorescently tagged endogenous FIB-1 with wrmScarlet in our tagged NUCL-1 strain. We find that NUCL-1 and FIB-1 occupy distinct nucleolar spaces in germ cells. FIB-1 forms a "net-like" structure that is filled in and surrounded by NUCL-1 (Fig. 2e). 3-dimensional reconstruction and volume rendering of a single nucleolus demonstrates this 2-compartment structure (Supplementary Videos 1 and 2).

We next wanted to determine the sub-nucleolar patterning of Y66H1A.4 and LPD-6, two RG/RGG proteins with predicted, but untested, nucleolar localization in *C. elegans*. Y66H1A.4 is a largely uncharacterized protein identified in our RG/RGG survey as containing two of the top 50 longest RG/RGG repeats in *C. elegans*. HHpred and MARRVEL toolkits predict shared homology with GAR1, so we named it GARR-1 (GAR1 related). *C. elegans* GARR-1 contains an N-terminal 48 amino acid-long repeat and a C-terminal 70 amino acid-long repeat (Fig. 1d). Both repeats are phenylalanine-rich, although only the C-terminal repeat contains the nucleolar sequence motif found in NUCL-1, FIB-1, and LPD-6. LPD-6 contains the 4th-longest RG/RGG repeat in *C. elegans* and is an ortholog of human PPAN-P2RY11 and yeast SSF1, nucleolar proteins required for ribosomal large subunit maturation[48–50]. As with NUCL-1, sGFP11 was placed at the C terminus of endogenous GARR-1 and LPD-6 for germline-specific visualization.

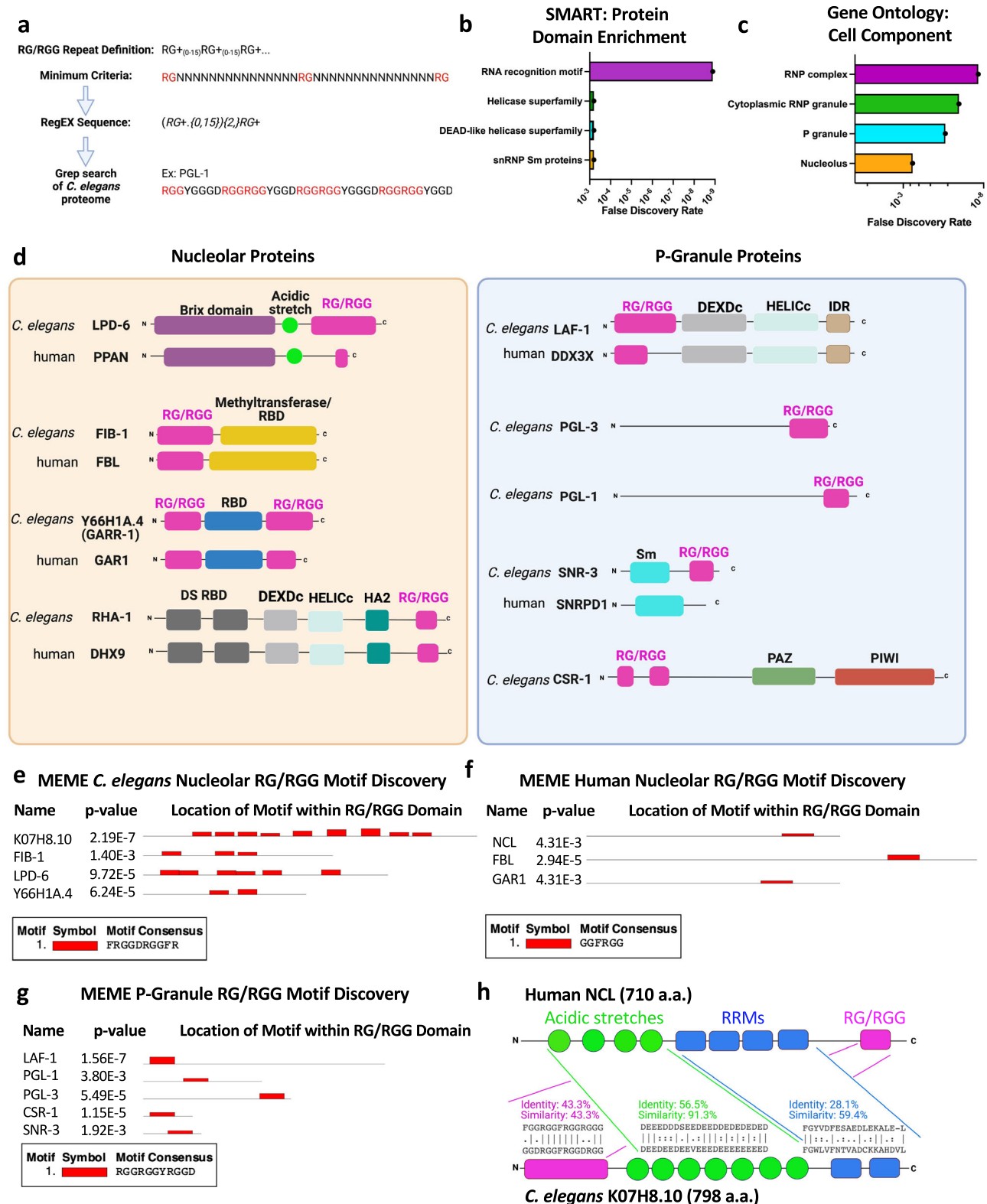

**Fig. 1 | RG/RGG-repeat survey in *C. elegans* reveals K07H8.10 as a homolog of human Nucleolin. a** Strategy for identifying regularly spaced RG/RGG repeats. **b** SMART protein domain enrichment of the 43 proteins that contain the 50 longest RG/RGG repeats. The 4 domains with the lowest false discovery rate are shown. The RRM is the most enriched domain in RG/RGG-containing proteins. **c** The 50 longest RG/RGG repeats are enriched in BMCs. **d** *C. elegans* Nucleolar and P-granule RG/RGG proteins and their human homologs. **e** MEME reveals a nucleolar RG/RGG motif. **f** A phenylalanine-rich motif is also present within RG/RGG repeats of human nucleolar proteins. **g** MEME motif discovery within the RG/RGG domains of P-granule proteins reveals a common tyrosine-rich motif. **h** Domain and pairwise sequence alignment between NCL and K07H8.10.

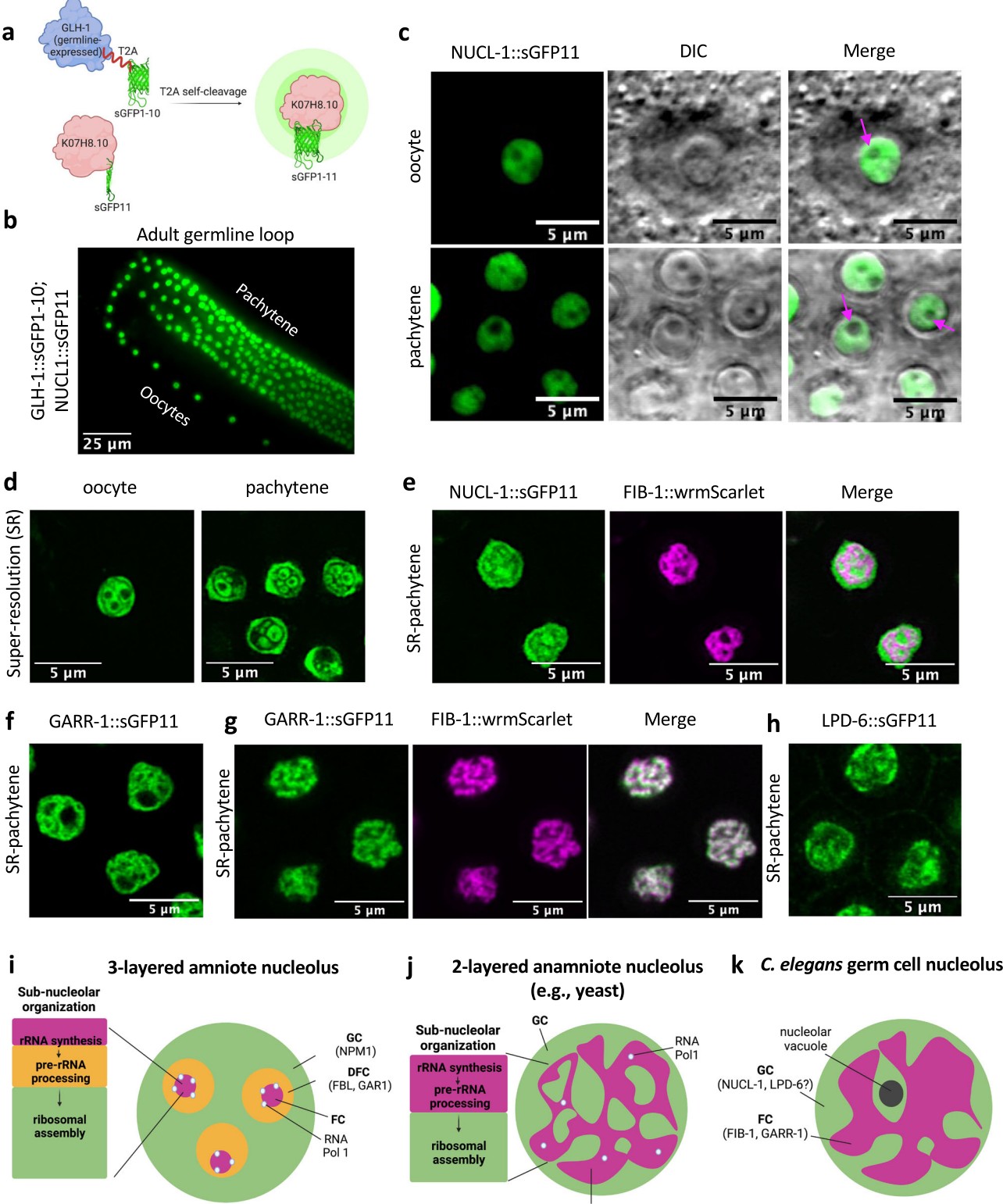

**Fig. 2 | NUCL-1 localizes to germ cell nucleoli and displays precise sub-nucleolar compartmentalization. a** Split-GFP strategy for K07H8.10 (NUCL-1) visualization in the germline. **b** NUCL-1 localizes to nucleoli in the adult germline. **c** Confocal-DIC imaging reveals that NUCL-1 localizes exclusively to the nucleolus in oocytes and pachytene germ cells. Pink arrows indicate nucleolar vacuoles. **d** NUCL-1 displays precise sub-nucleolar organization within oocytes (1 nucleolus shown) and pachytene germ cells (5 nucleoli shown). **e** NUCL-1 and FIB-1 occupy distinct spaces in pachytene germ cell nucleoli. **f** GARR-1::sGFP11 in pachytene germ cell nucleoli.

**g** GARR-1 and FIB-1 occupy the same sub-nucleolar compartment in pachytene germ cells. **h** LPD-6::sGFP11 in pachytene germ cells. **i** A human nucleolus is organized into 3 phase-separated layers and ribosome biogenesis occurs in a step-by-step manner from the inside-out. **j** Yeast nucleoli are organized into 2 phase-separated compartments. **k** *C. elegans* nucleoli contain 2 phase-separated compartments, with NUCL-1 and LPD-6 in the GC and FIB-1 and GARR-1 in the FC. (FC fibrillar center, DFC dense fibrillar component, GC granular component, RNA Pol1 RNA polymerase 1).

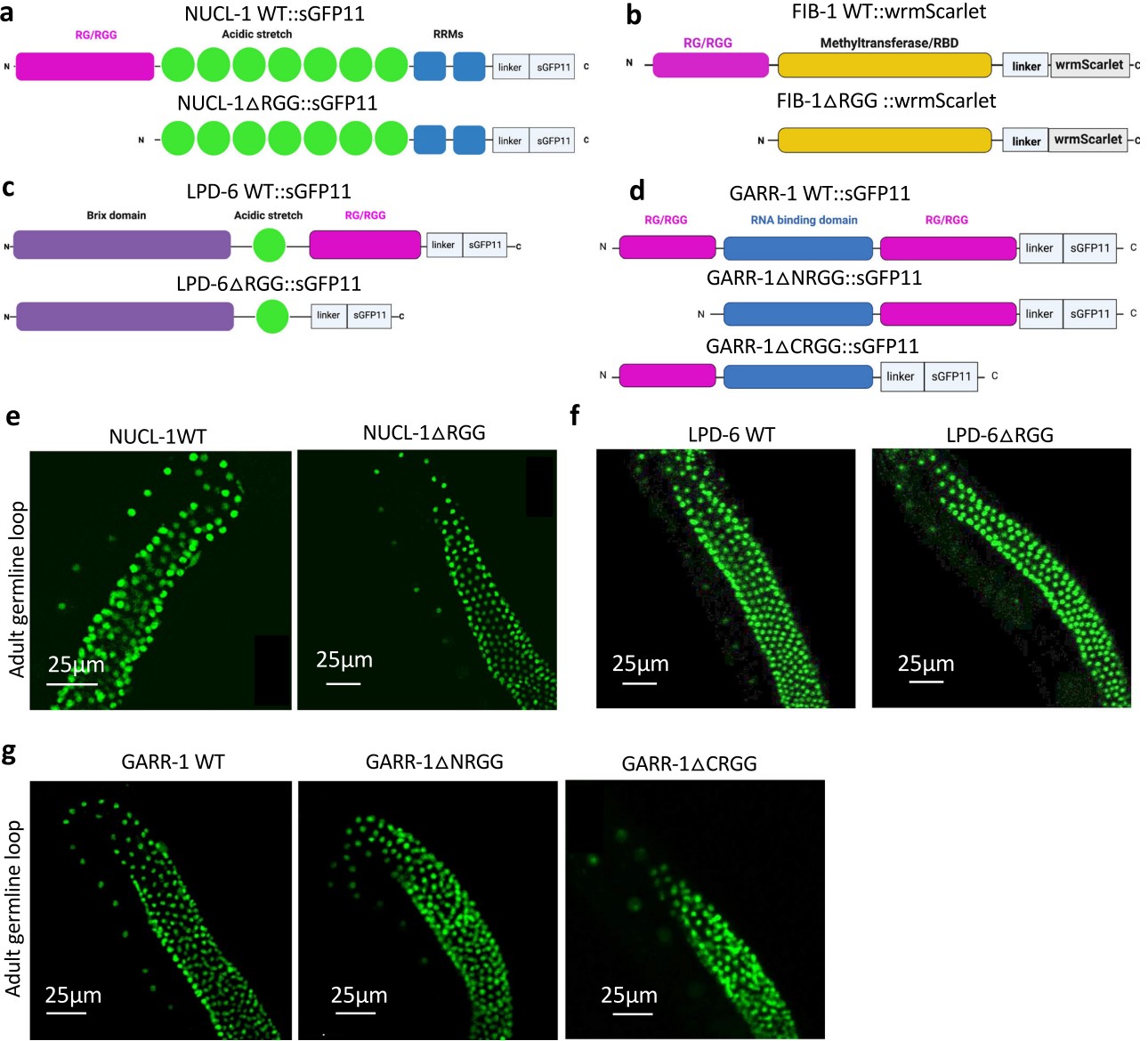

**Fig. 3 | NUCL-1, GARR-1, and LPD-6 RG/RGG domains are not required for nucleolar accumulation. a** sGFP11-tagged NUCL-1 and the RG/RGG domain deletion. **b** wrmScarlet-tagged FIB-1 and the RG/RGG domain deletion. **c** sGFP11-tagged LPD-6 and the RG/RGG domain deletion. **d** sGFP11-tagged GARR-1 and the N- and C-terminal RG/RGG domain deletions. **e** NUCL-1ΔRGG accumulates in germline nucleoli. **f** LPD-6ΔRGG accumulates in germ cell nucleoli. **g** GARR-1ΔNRGG and GARR-1ΔCRGG accumulate in germline nucleoli.

Both proteins accumulate in germ cell nucleoli at all stages of worm development. GARR-1 displays a "net-like" structure similar to FIB-1 (Fig. 2f). Crossing our tagged GARR-1 line into FIB-1::wrmScarlet worms revealed that GARR-1 and FIB-1 occupy the same nucleolar sub-compartment (Fig. 2g). Compared to other nucleolar proteins, the compartmentalization of LPD-6 is not as clear. However, its enrichment both at the outer edge and in the interior of nucleoli appears most similar to NUCL-1 (Fig. 2h).

Nucleoli of amniotes are organized into three phase-separated layers; the innermost fibrillar center (FC), the middle dense fibrillar component (DFC) and the outer granular component (GC). rRNA transcription occurs at the border between the FC and DFC, rRNA is chemically modified in the DFC, and ribosomal subunits are assembled in the GC. RNA polymerase 1 is often used as a marker of the FC, FBL for the DFC, and Nucleophosmin for the GC (Fig. 2i)[3]. This 3-layered structure is thought to facilitate the step-by-step process of ribosome production from the inside out. In contrast, anamniotes (i.e., lower eukaryotes) contain a 2-compartment nucleolus[51]. RNA polymerase 1 is

scattered in spots throughout the FC, which houses both rRNA transcription and modification. The GC surrounds and penetrates through the FC as it accepts rRNA ready for higher-order assembly (Fig. 2j). *C. elegans* germ cell nucleoli more closely reflect a two-layer organization, with FIB-1 and GARR-1 residing in the FC and NUCL-1 and LPD-6 residing in the GC (Fig. 2k). Our super-resolution images provide the first visualization of *C. elegans* sub-nucleolar structure in live worms.

## RG/RGG domains are dispensable for nucleolar accumulation
To test if RG/RGG domains are required for nucleolar accumulation in *C. elegans*, we deleted the RG/RGG repeats from the tagged NUCL-1, FIB-1, GARR-1, and LPD-6 strains (Fig. 3a–d, Supplementary Fig. 3). In mice, homozygous NCL RG/RGG deletion is not viable[29]. In *C. elegans*, NUCL-1, GARR-1, and LPD-6 RG/RGG deletions are viable as homozygotes, providing tools for future in vivo study of the domain. Fibrillarin knockout is lethal in *S. cerevisiae* and mouse, and in *C. elegans* we find that full FIB-1 RG/RGG deletion does not provide viable worms[52,53]. In addition, throughout two rounds of CRISPR injections we

were unsuccessful at deleting a second GARR-1 RG/RGG domain when the first had been deleted. This suggests that worms missing both GARR-1 RG/RGG domains may not be viable.

Despite containing a signature nucleolar motif, we find that NUCL-1, GARR-1, and LPD-6 RG/RGG repeats are completely dispensable for nucleolar accumulation in germ cells. NUCL-1ΔRGG, GARR-1ΔNRGG, GARR-1ΔCRGG, and LPD-6ΔRGG proteins localize to germ cell nucleoli at all stages of worm development (Fig. 3e–g). Based on GFP intensity of pachytene nucleoli, all ΔRGG proteins were expressed in the germline at WT or near-WT levels (Supplementary Fig. 3).

## RG/RGG domains are required for sub-nucleolar compartmentalization

Despite the normal nucleolar accumulation of RG/RGG deletion proteins, live super-resolution confocal imaging of pachytene germ cell nucleoli revealed a disruption of sub-nucleolar organization. WT NUCL-1 displays clear regions of enrichment and depletion, but NUCL-1ΔRGG lacks this compartmentalization (Fig. 4a, a'). A histogram of GFP intensity within a single WT NUCL-1 nucleolus shows a biphasic distribution (Fig. 4b, c). In contrast, NUCL-1ΔRGG is more homogeneously dispersed throughout nucleoli, and fluorescence intensity falls into a single peak (Fig. 4b', c'). To quantify NUCL-1 homogeneity within individual nucleoli, the coefficient of variation (CV) was calculated by dividing the standard deviation by the mean intensity of fluorescence. A high CV indicates heterogeneous distribution, while a low CV indicates homogeneous distribution. The CV is decreased in NUCL-1ΔRGG worms compared to WT, indicating the more homogeneous distribution of NUCL-1 within nucleoli and loss of sub-nucleolar organization (Fig. 4d). Nucleolar size and morphology are closely associated with the metabolic state of cells[54–56]. As a loose proxy for nucleolar function, the volume of labeled NUCL-1 and NUCL-1ΔRGG was quantified. No difference in the volume of nucleoli was observed between WT and NUCL-1ΔRGG, suggesting that loss of the RG/RGG domain and NUCL-1's subsequent redistribution in the nucleolus may be inconsequential to germ cell metabolism (Fig. 4e).

To determine if the compartmentalization of other nucleolar proteins is disrupted when the NUCL-1 RG/RGG domain is absent, we again tagged endogenous FIB-1 at its C-terminus with full-length wrmScarlet in the NUCL-1ΔRGG strain. FIB-1 still localized to germ cell nucleoli and was expressed at similar levels in both WT and NUCL-1ΔRGG worms (Fig. 4f and Supplementary Fig. 3). In NUCL-1ΔRGG worms, FIB-1 lost its distinct compartmentalization into the FC and was more homogeneously distributed throughout nucleoli; without a change in nucleolar volume (Fig. 4f–h). Although NUCL-1 and FIB-1 lose large-scale compartmentalization in NUCL-1ΔRGG nucleoli, their localization is still partially non-overlapping. A profile plot of fluorescence across a single WT nucleolus clearly demonstrates the opposite localization patterns of NUCL-1 and FIB-1 (Supplementary Fig. 4). A profile plot across a NUCL-1ΔRGG nucleolus reveals that, while more homogenously distributed, the NUCL-1ΔRGG and FIB-1 proteins still show distinct patterns of expression. Where NUCL-1ΔRGG expression is highest, FIB-1 expression is still lowest and where NUCL-1ΔRGG expression is lowest, FIB-1 expression is still highest (Supplementary Fig. 4).

Although our split GFP lines do not allow for visualization of NUCL-1 itself outside of the germline, we wondered if the NUCL-1 RG/RGG domain is important for sub-nucleolar organization of FIB-1 in somatic cells. We chose to image FIB-1 in nucleoli of hypodermal cells, which align along the top of the worm and are readily identifiable (Fig. 4i). In hypodermal nucleoli FIB-1 displays a similar "net-like" organization to what is observed in germ cells. However, FIB-1 loses this organization when the NUCL-1 RG/RGG domain is absent (Fig. 4j). FIB-1 displays a significantly decreased CV compared to FIB-1 in WT nucleoli without a change in volume (Fig. 4k, l). These results indicate that the NUCL-1 RG/RGG domain is required for both fibrillar and granular compartmental specificity within germline and somatic nucleoli.

We next tested if GARR-1 or LPD-6 RG/RGG domains function similarly to the NUCL-1 RG/RGG domain in sub-nucleolar compartmentalization. Super-resolution imaging revealed that deletion of either the N- or C-terminal GARR-1 RG/RGG domain impaired organization of GARR-1 into the FC, resulting in a more homogeneous distribution of the protein within nucleoli (Fig. 5a, b). In contrast to highly compartmentalized proteins such as NUCL-1, FIB-1, and GARR-1, LPD-6 shows less clear sub-nucleolar compartmentalization. However, LPD-6 has a significantly higher CV compared to free GFP, indicating some degree of organization. Removal of the LPD-6 RG/RGG domain does not disrupt this organization, but does increase its variability (Fig. 5c, d). In summary, these data show that multiple nucleolar RG/RGG domains are dispensable for nucleolar accumulation in *C. elegans* germ cells, and instead show a role in the phase separation of proteins into nucleolar sub-compartments.

## Genetic and functional interaction of NUCL-1 and FIB-1 RG/RGG domains

Because FIB-1 compartmentalization depends on the NUCL-1 RG/RGG domain, we asked if the NUCL-1 and FIB-1 RG/RGG domains display a genetic interaction. To test this, we deleted the RG/RGG domain of endogenous FIB-1 on both WT NUCL-1 and NUCL-1ΔRGG backgrounds (Fig. 6a). As previously stated, homozygous deletion of the FIB-1 RG/RGG domain is not viable. Only heterozygous FIB-1 RG/RGG deletion worms were recovered, which are viable on the WT NUCL-1 background but not viable on the NUCL-1ΔRGG background past the F1 generation. Maternal contribution of WT FIB-1 allowed for F1 heterozygotes in the NUCL-1ΔRGG background, but lack of a full complement of WT FIB-1 in subsequent generations was lethal only when the NUCL-1 RG/RGG domain was absent. These results support a partial redundancy of the NUCL-1 and FIB-1 RG/RGG repeat domains, likely stemming from their similar amino acid composition. These results also suggest that the FIB-1 RG/RGG domain could be capable of rescuing sub-nucleolar organization of NUCL-1 in ΔRGG worms.

To determine if the FIB-1 RG/RGG domain and the NUCL-1 RG/RGG domain are exchangeable, we placed the full FIB-1 RG/RGG domain onto the N-terminus of tagged NUCL-1ΔRGG worms (Fig. 6b and Supplementary Fig. 3). The NUCL-1(FIB-1RGG) protein accumulated in germ cell nucleoli and showed subtly improved sub-nucleolar organization (Fig. 6c–e). However, the incomplete rescue may be due to lower expression of NUCL-1(FIB-1RGG), as GFP expression in pachytene nucleoli was decreased by 85% and 82% compared to WT and NUCL-1ΔRGG, respectively (Supplementary Fig. 3). How addition of the FIB-1 RG/RGG domain causes decreased levels of NUCL-1 is unclear but could be explained by increased protein degradation due to abnormal protein folding. Because NCL is regulated both at the level of transcription and translation, it is also possible that the FIB-1 RG/RGG domain may take on a gain-of-function role that inhibits expression at either level[57,58]. Aside from expression level, the NUCL-1 RG/RGG domain is 40% longer than the FIB-1 RG/RGG domain, so the lack of a complete rescue could also be explained by a smaller capacity of the FIB-1 RG/RGG for protein and RNA interactions within the nucleolus.

## The effects of RG/RGG domain deletion on fertility and development

NUCL-1ΔRGG germ cells show sub-nucleolar disorganization of both NUCL-1 and FIB-1. Despite this disorganization, NUCL-1ΔRGG worms are fertile with WT broods; even at the sub-optimal temperatures of 15 °C and 25 °C, and up to the 80th generation post-editing (Fig. 7a). NUCL-1ΔRGG somatic cells show sub-nucleolar disorganization of FIB-1. However, NUCL-1ΔRGG worms show no significant difference in time to adulthood compared to WT, indicating normal somatic

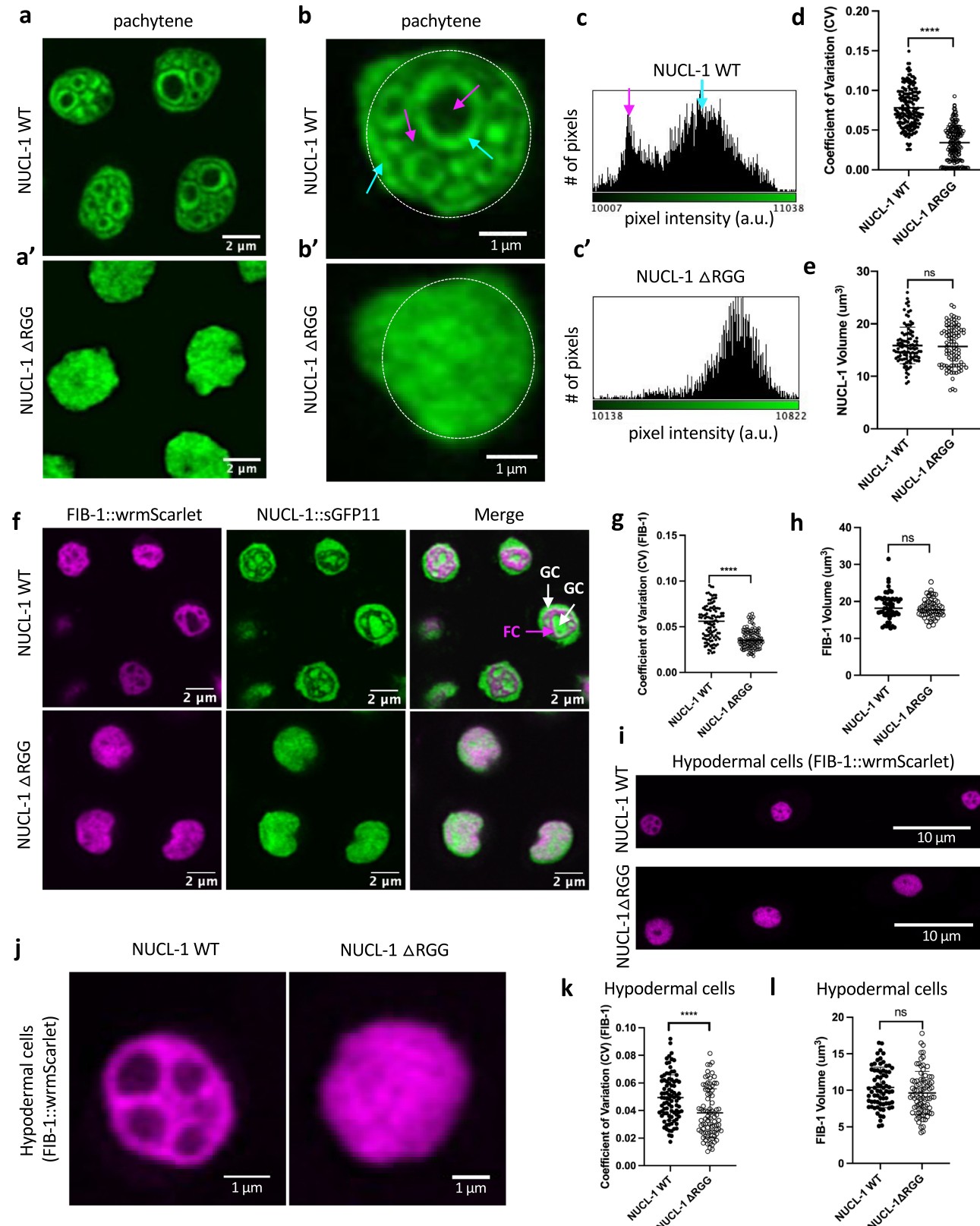

**Nature Communications** | (2022)13:6585

development (Fig. 7b). Similarly, LPD-6ΔRGG worms show no decrease in brood size (Fig. 7c). In contrast, both the N- and C-terminal GARR-1 RG/RGG domains are crucial for fertility and development. Deleting either domain results in decreased brood size at 20 °C, which is not enhanced at 25 °C. However, at 15 °C GARR-1ΔNRGG and GARR-1ΔCRGG worms produce no viable progeny, perhaps because loss of the RG/RGG domain results in impaired condensate dynamics at

colder temperatures (Fig. 7d). When the fertility of adult worms is scored by looking for fertilized embryos in the uterus, GARR-1ΔCRGG worms show modest decreases in percent fertility at 20 °C and 25 °C, while both GARR-1ΔNRGG and GARR-1ΔCRGG worms are sterile at 15 °C (Fig. 7e). Both GARR-1ΔNRGG and GARR-1ΔCRGG worms show delayed somatic development (Fig. 7f). WT GARR-1 worms take 54.4 ± 1.3 h to reach maturity from the first larval stage, while GARR-1ΔNRGG

**Fig. 4 | The NUCL-1 RG/RGG domain contributes to sub-nucleolar organization in the germline and in the soma. a** Pachytene germ cell nucleoli in NUCL-1 WT and **a'** NUCL-1ΔRGG adult worms (4 nucleoli per image). **b** Single nucleoli in NUCL-1 WT and **b'** NUCL-1ΔRGG pachytene germ cells. White dashed line represents area measured. NUCL-1 WT is enriched in the GC (blue arrows) and depleted in the FC and nucleolar vacuoles (pink arrows). **c** Histogram of fluorescence intensity for the nucleolus in **b**. NUCL-1 WT intensities show peaks corresponding to areas of enrichment (blue arrow) and depletion (pink arrow). **c'** Histogram of fluorescence intensity for the nucleolus in **b'**. NUCL-1ΔRGG intensities group into a single peak. **d** NUCL-1ΔRGG CV is decreased compared to WT (*n* = 168 nucleoli from 12 NUCL-1 WT worms, 183 nucleoli from 12 NUCL-1ΔRGG worms). *p* = 2.67E-51. **e** No difference in nucleolar volume was detected in NUCL-1ΔRGG worms compared to WT (*n* = 100 nucleoli from 10 NUCL-1 WT worms and 91 nucleoli from 10 NUCL-1ΔRGG worms). **f** FIB-1::wrmScarlet and NUCL-1::sGFP11 in NUCL-1 WT and NUCL-1ΔRGG pachytene germ cell nucleoli. In NUCL-1 WT nucleoli, FIB-1 is in the FC (pink arrow) and NUCL-1

is in the GC (white arrows). FIB-1 in NUCL-1ΔRGG nucleoli shows no compartmentalization into the FC. **g** FIB-1 in NUCL-1ΔRGG nucleoli has decreased CV compared to WT (*n* = 111 nucleoli from 7 NUCL-1 WT worms and 118 nucleoli from 7 NUCL-1ΔRGG worms). *p* = 3.19E-16. **h** The volume of FIB-1-labeled nucleoli in NUCL-1ΔRGG worms is comparable to WT (*n* = 55 nucleoli from 7 NUCL-1 WT worms and 61 nucleoli from 7 NUCL-1ΔRGG worms). **i** FIB-1 in NUCL-1WT and NUCL-1ΔRGG hypodermal cells. 3 nucleoli shown per image. **j** Single FIB-1-labeled nucleoli in NUCL-1WT and NUCL-1ΔRGG hypodermal cells. **k** The CV for FIB-1 in NUCL-1ΔRGG hypodermal nucleoli is decreased compared to WT (*n* = 88 nucleoli from 10 NUCL-1 WT worms and 84 nucleoli from 11 NUCL-1ΔRGG worms). *p* = 2.89E-05. **l** The volume of FIB-1-labeled hypodermal cell nucleoli in NUCL-1ΔRGG worms is comparable to WT (*n* = 70 nucleoli from 9 NUCL-1 WT worms and 81 nucleoli from 10 NUCL-1ΔRGG worms). **d**, **e**, **g**, **h**, **k**, **l** Unpaired, two-tailed *t* test with Welch's correction. Points on graphs represent individual nucleoli and means ± SD are shown.

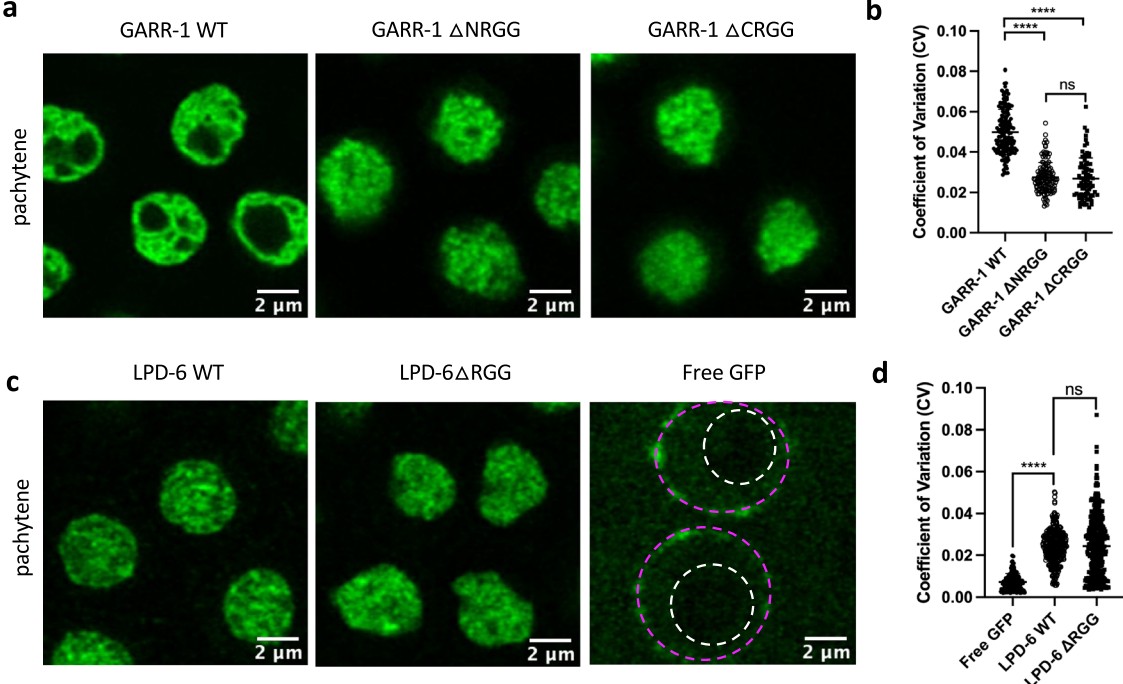

**Fig. 5 | The GARR-1 N and C terminal RG/RGG domains contribute to sub-nucleolar organization in the germline. a** GARR-1 WT localizes to the FC of pachytene germ cell nucleoli. GARR-1ΔNRGG and GARR-1ΔCRGG show a loss of this compartmentalization **b** and decreased CV compared to WT (*n* = 152 nucleoli from 8 GARR-1 WT worms, 135 nucleoli from 8 GARR-1ΔNRGG worms, and 84 nucleoli from 6 GARR-1ΔCRGG worms). ****p* < 0.0001. **c** LPD-6ΔRGG protein does not show differences in nucleolar organization compared to WT. Both LPD-6 WT and LPD-

6ΔRGG are less homogenously dispersed than free GFP. **d** There is no significant difference in CV between LPD-6 and LPD-6ΔRGG, but both have a significantly greater CV compared to free GFP (*n* = 351 nucleoli from 14 LPD-6 WT worms, 335 nucleoli from 14 LPD-6ΔRGG worms, and 159 nucleoli from nine DUP218 (GLH-1::T2A::sGFP1-11) worms). ****p* < 0.001. **b**, **d** One-way ANOVA with Tukey's multiple comparisons test. Points represent individual nucleoli and means ± SD are shown.

worms take 62.3 ± 2.0 h and GARR-1ΔCRGG worms take 65.8 ± 2.3 h. Embryonic hatching is also incomplete at 18 h post-laying but comparable to WT by 24 h. (Fig. 7g).

**Full deletion of NUCL-1 produces viable worms**
Given NUCL-1's role in nucleolar organization, we next wanted to determine if NUCL-1 is an essential gene in *C. elegans*. We used CRISPR/Cas9 genome editing to delete the entire coding sequence in the tagged strain (Supplementary Fig. 5). We then performed embryonic hatching and brood size assays to evaluate fertility, and a vulval maturation assay to measure somatic development. These assays were performed at both early and late generations post-editing because mutations in NUCL-1 could cause multi-generational phenotypes. In *C. elegans*, disruptions in the maintenance of genome integrity commonly result in the "mortal germline phenotype," a progressive loss of fertility over many generations[59]. At both 5 and 40

generations post-editing, worms lacking NUCL-1 (ΔNUCL-1) are viable, with the percentage of embryos hatched comparable to wild-type (WT) (Supplementary Fig. 5). ΔNUCL-1 worms also produce progeny at numbers similar to WT, but demonstrate a slight sensitivity to temperature (Fig. 7h, Supplementary Fig. 5). However, somatic development of ΔNUCL-1 worms is delayed. WT worms take 53.5 ± 1.5 h to develop from the first larval stage to a young adult with a fully mature vulva. In contrast, ΔNUCL-1 worms at 5 generations post-editing take 63.3 ± 1.7 h, and ΔNUCL-1 worms at 40 generations post-editing take 63.9 ± 1.4 h to fully develop (Fig. 7i). Thus, the lack of NUCL-1 has minimal effects on fertility and embryonic viability, but instead results in delayed development, suggestive of a reduced rate of protein synthesis or a possible heterochronic defect. Deleting NCL in other animal models is not viable, and *C. elegans* provides a unique opportunity to study NCL function and mechanisms of NCL-driven development.

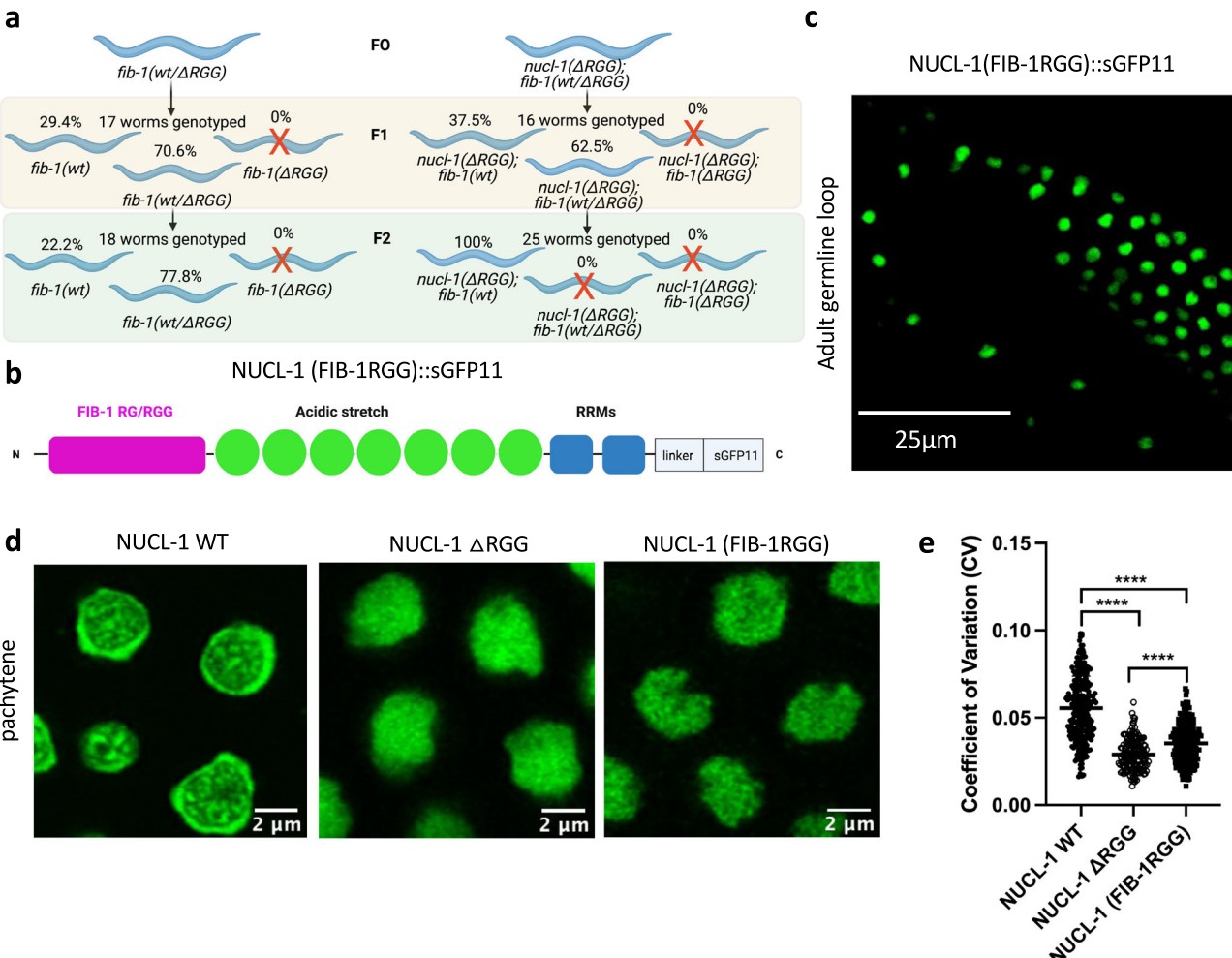

**Fig. 6 | The FIB-1 RG/RGG domain shares partial genetic and functional redundancy with the NUCL-1 RG/RGG domain. a** Deletion of FIB-1 RGG on NUCL-1 WT (left) and NUCL-1ΔRGG (right) backgrounds. Red X's represent non-viability. **b** NUCL-1(FIB-1RGG)::sGFP11 protein. **c** NUCL-1(FIB-1RGG) localizes to germ cell nucleoli. **d** Pachytene germ cell nucleoli in NUCL-1 WT, NUCL-1ΔRGG, and NUCL-1(FIB-1RGG) worms. NUCL-1(FIB-1RGG) protein lacks compartmentalization into the GC and **e** only shows a slight improvement in CV compared to NUCL-1ΔRGG (*n* = 251 nucleoli from 13 NUCL-1 WT worms, 169 nucleoli from 9 NUCL-1ΔRGG worms, and 209 nucleoli from 12 NUCL-1(FIB-1RGG) worms). ****$p < 0.0001$; One-way ANOVA with Tukey's multiple comparisons test. Points represent individual nucleoli and means ± SD are shown.

Although no paralogs of NUCL-1 were found in our bioinformatic analysis of RG/RGG domains, other RG/RGG nucleolar proteins may compensate for the loss of NUCL-1. The LPD-6 RG/RGG domain is 127 amino acids long and second in length only to NUCL-1. Because LPD-6 appears to reside in the GC with NUCL-1, we hypothesized that the LPD-6 RG/RGG domain may buffer the effects of NUCL-1 deletion. To test for a functional relationship between NUCL-1 and the LPD-6 RG/RGG domain, we crossed ΔNUCL-1 worms with LPD-6ΔRGG worms and performed a brood size assay. Neither ΔNUCL-1 nor LPD-6ΔRGG worms show a difference in brood size compared to WT, but ΔNUCL-1;LPD-6ΔRGG worms show a 37% decrease in the number of progeny per worm (Fig. 7j). Thus, NUCL-1 deletion is viable in nematodes likely through compensation by other nucleolar RG/RGG proteins, including LPD-6.

## Discussion

An important question in BMC biology is how phase separation into a specific BMC or sub-condensate is encoded into proteins. Our study provides four examples of nucleolar RG/RGG domains that are dispensable for in vivo phase separation into nucleoli. Instead, three of these domains function in sub-nucleolar phase separation (Fig. 8a). The N and C-terminal GARR-1 RG/RGG domains compartmentalize GARR-1 into the FC, while the NUCL-1 RG/RGG domain is required for NUCL-1 organization into the GC and FIB-1 organization into the FC. As the longest nucleolar RG/RGG repeat in *C. elegans*, the NUCL-1 RG/RGG domain may be responsible for scaffolding of both the FC and the GC. Each RG/RGG domain that functions in sub-nucleolar phase separation is phenylalanine-rich, and both the NUCL-1 and GARR-1 C-terminal RG/RGG domains contain the common phenylalanine-rich motif. However, the LPD-6 RG/RGG domain also contains this motif and has no role in sub-nucleolar organization. Furthermore, the RG/RGG protein FUST-1 (FUS/TLS RNA binding protein homolog) contains both a nucleolar-like and a P-granule-like RG/RGG repeat. FUST-1 shows diffuse nuclear localization, but in germ cells localizes to P-granules[60]. Whether or not the phenylalanine-rich nucleolar motif is essential for sub-nucleolar phase separation, and which additional RG/RGG features may contribute, will be the subject of future study.

A second important question in BMC biology is how multi-layered structure facilitates function. Our data demonstrate that removal of RG/RGG domains from either NUCL-1 or GARR-1 disrupt germ cell sub-nucleolar structure, yet only GARR-1 RG/RGG removal results in fertility defects. Although large-scale compartmentalization is certainly disrupted when the NUCL-1 RG/RGG domain is absent, it is possible that FC and GC layers are still present, although

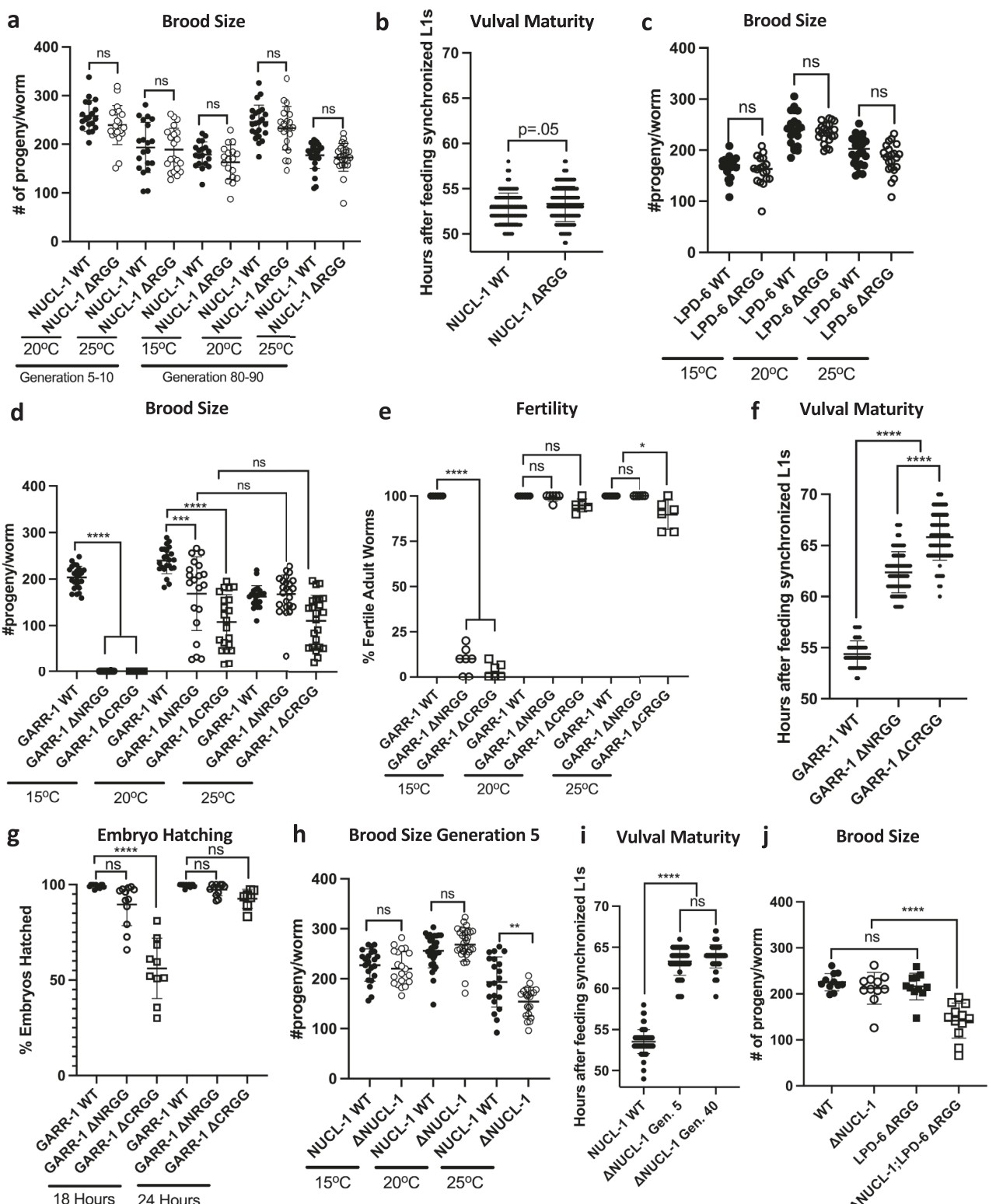

at a much finer scale. In NUCL-1ΔRGG nucleoli, NUCL-1 and FIB-1 dispersal is not fully homogeneous, and localization of both proteins is still partially non-overlapping (Supplementary Videos 3 and 4). Thus, ribosome biogenesis may not be affected. On the other hand, NUCL-1ΔRGG worms may have defects in ribosome biogenesis and translation below the threshold for impacting fertility. The differences in fertility and viability when deleting the RG/RGG domain from NUCL-1 compared to GARR-1 and FIB-1 may also come down to

protein function. Both GARR-1 and FIB-1 are members of snoRNPs and their RG/RGG domains may also function in assembly of these essential BMCs. Regardless, our data decouple precise, large-scale sub-nucleolar phase separation with nucleolar function and bring into question the role of canonical nucleolar sub-compartmentalization in ribosome biogenesis. As recently suggested, it may be that "latent" ribosome assembly factors, or those not actively participating in ribosome biogenesis, form phase

**Fig. 7 | The effects of RG/RGG domain deletion on fertility and development.**
**a** NUCL -1ΔRGG worms show no defect in brood size (Generation 5–10: 20 °C n = 19 WT and 21 ΔRGG worms, 25 °C n = 20 WT and 21 ΔRGG worms. Generation 80–90: 15 °C n = 19 WT, 18 ΔRGG worms, 20 °C n = 23 WT and 22 ΔRGG worms, 25 °C n = 25 WT and 24 ΔRGG worms. **b** NUCL-1 ΔRGG worms have normal vulval maturity (n = 103 WT and 134 ΔRGG worms). **c** LPD-6ΔRGG worms show no defect in brood size (15 °C: n = 20 WT and 18 ΔRGG worms, 20 °C n = 22 WT and 22 ΔRGG worms, 25 °C n = 24 WT and 22 ΔRGG worms). **d** GARR-1ΔNRGG and GARR-1ΔCRGG worms have a smaller brood size at 20 °C than GARR-1 WT and produce no progeny at 15 °C (15 °C n = 24 WT, ΔNRGG and ΔCRGG worms, 20 °C n = 23 WT, 20 ΔNRGG, and 21 ΔCRGG worms, 25 °C n = 23 WT, 24 ΔNRGG, and 23 ΔCRGG worms). ****p = <0.0001. **e** Percentage of fertile worms is decreased in GARR-1ΔNRGG and GARR-1ΔCRGG, especially at 15 °C (15 °C: n = 6 WT, 7 ΔNRGG, and 6 ΔCRGG; 20 °C: n = 6 WT, 6 ΔNRGG, and 5 ΔCRGG; 25 °C: n = 7 WT, 6 ΔNRGG, and 6 ΔCRGG experimental replicates of 20 worms each). 15 °C ****p < 0.0001; 25 °C GARR-1WTvs GARR-

1ΔCRGG p = 0.0016, GARR-1ΔNRGG vs GARR-1ΔCRGG p = 0.0026. **f** GARR-1ΔNRGG and GARR-1ΔCRGG worms have delayed vulval maturation (n = 46 WT, 107 ΔNRGG, and 83 ΔCRGG worms). ****p = <0.0001. **g** GARR-1ΔNRGG and GARR-1ΔCRGG embryos show delayed hatching (18 h: n = 12 WT, 12 ΔNRGG, and 10 ΔCRGG; 24 h: n = 12 WT, 12 ΔNRGG, and 7 ΔCRGG experimental replicates of 20 worms each). ****p = <0.0001. **h** 5 generations after NUCL-1 deletion worms have a subtly decreased brood size at 25 °C (15 °C: n = 22 WT and 21 ΔNUCL-1 worms, 20 °C n = 30 WT and 30 ΔNUCL-1 worms, 25 °C n = 21 WT and 21 ΔNUCL-1 worms). p = 0.0064. **i** ΔNUCL-1 worms show delayed vulval maturation at 20 °C (n = 64 WT, 55 5th Gen. ΔNUCL-1, and 64 40th Gen. ΔNUCL-1 worms). ****p = <0.0001. **j** Although ΔNUCL-1 and LPD-6ΔRGG worms have normal brood sizes, double mutant ΔNUCL-1; LPD-6ΔRGG worms produce significantly fewer progeny (n = 11 WT, 11 ΔNUCL-1, 11 LPD-6 ΔRGG, and 12 ΔNUCL-1; LPD-6ΔRGG worms). ****p = <0.0001. **a, c–j** One-way ANOVA with Tukey's multiple comparisons test. **b** Unpaired, two-tailed t-test with Welch's correction. All data are shown as means ± SD.

## a

**Summary of Nucleolar RG/RGG Domains**

| RG/RGG Domain | F-rich Motif? | Nucleolar Subcompartment | Required for nucleolar accumulation? | Required for sub-nucleolar organization? | Phenotype of ΔRG/RGG worms |
|---|---|---|---|---|---|
| **NUCL-1** | Yes | GC | No | Yes- NUCL-1 & FIB-1 | Fully viable and fertile<br>Normal development |
| **FIB-1** | Yes | FC | Unknown | Unknown | Not viable |
| **GARR-1** | Yes | FC | Nterm RGG- No | Yes- GARR-1 | Viable with decreased fertility and cold-induced sterility<br>Delayed somatic development |
| | | | Cterm RGG- No | Yes- GARR-1 | Viable with decreased fertility and cold-induced sterility<br>Delayed somatic development |
| **LPD-6** | Yes | GC | No | No | Fully viable and fertile |

## b

Wild-type RG/RGG

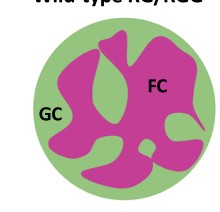

Below Critical RG/RGG Threshold?

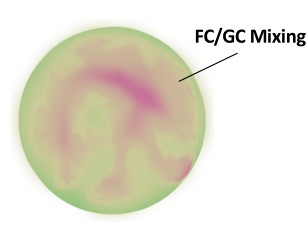

FC/GC Mixing

**Fig. 8 | RG/RGG repeat function in the *C. elegans* nucleolus. a** Summary chart of nucleolar RG/RGG domain localization and functions. **b** RG/RGG domains are critical for large-scale compartmentalization of FC and GC regions in *C. elegans* nucleoli. Loss of certain RG/RGG domains (NUCL-1, GARR-1) may bring the concentration below a critical threshold at which nucleolar sub-compartments can organize. Below this threshold, some small-scale organization of FC and GC regions may still occur (FC fibrillar center, GC granular component).

separated sub-condensates, while the function of actively engaged factors is not dependent upon such organization[13].

Finally, the functional redundancy between the NUCL-1 and FIB-1 RG/RGG domains and LPD-6 and NUCL-1 may indicate the need for a critical mass of RG/RGG repeats to support normal nucleolar structure and function. Perhaps the difference between the compartmentalization-supporting RG/RGG domains (NUCL-1 and GARR-1) and the LPD-6 RG/RGG domain comes down to levels of expression. In the *C. elegans* germline the NUCL-1 and GARR-1 transcripts are abundant (435.9 FPKM and 326.29 FPKM) while the LPD-6 transcript is not (40.63 FPKM)[32]. Thus, removing the LPD-6 RG/RGG domain may not decrease the amount of RG/RGG repeats enough to affect sub-nucleolar compartmentalization of any protein. In contrast, removal of the NUCL-1 or GARR-1 RG/RGG domains may bring the overall amount of RG/RGG repeats below a critical threshold at which large-scale nucleolar sub-structure is disrupted (Fig. 8b). If so, fine-tuning the concentration of RG/RGG repeats may represent a new way to manipulate BMC substructure.

## Methods

### Strain generation and maintenance

*C. elegans* strains were maintained using standard protocols (Brenner 1974). Strains created for this study include **DUP229** glh-1(sam129[glh-1::T2A::sGFP2(1-10)]) I; nucl-1(sam132[NUCL-1::M3]) IV; **DUP239** glh-1(sam129[glh-1::T2A::sGFP2(1-10)]) I; nucl-1(sam142[NUCL-1(RGG deletion)::M3]) IV; **DUP242** glh-1(sam129[glh-1::T2A::sGFP2(1-10)]) I; nucl-

1(sam142[NUCL-1(RGG deletion)::M3]) IV; fib-1(sam144[fib-1::wrmScarlet]) V; **DUP243** glh-1(sam129[glh-1::T2A::sGFP2(1-10)]) I; nucl-1(sam132[NUCL-1::M3]) IV; fib-1(sam144[fib-1::wrmScarlet])V; **DUP244** glh-1(sam129[glh-1::T2A::sGFP2(1-10)]) I; nucl-1(sam132[NUCL-1::M3]) IV; fib-1(sam145[fib-1delRGG])V; **DUP245** glh-1(sam129[glh-1::T2A::sGFP2(1-10)]) I; nucl-1(sam142[NUCL-1(RGG deletion)::M3]) IV; fib-1(sam145[fib-1delRGG])V; **DUP248** glh-1(sam129[glh-1::T2A::sGFP2(1-10)]) I; nucl-1(sam147[NUCL-1(deletion)::M3]) IV; **DUP257** glh-1(sam129[glh-1::T2A::sGFP2(1-10)]) I; nucl-1(sam156[NUCL-1(FIB-1 RGG)::M3]) IV; **DUP249** glh-1(sam129 [glh-1::T2A::sGFP2(1-10)]) I; lpd-6(sam148[LPD-6::M3])I; **DUP252** glh-1(sam129[glh-1::T2A::sGFP2(1-10)]) I; lpd-6(sam151[LPD-6delRGG::M3])I; **DUP250** glh-1(sam129[glh-1::T2A::sGFP2(1-10)]) I; garr-1(sam149 [GARR-1::M3])IV; **DUP253** glh-1(sam129[glh-1::T2A::sGFP2(1-10)]) I; garr-1(sam152[GARR-1-delNtermRGG::M3])IV; **DUP260** glh-1(sam129 [glh-1::T2A::sGFP2(1-10)]) I; garr-1(sam158[GARR-1-delCtermRGG::M3])IV. **DUP264** glh-1(sam129[glh-1::T2A::sGFP2(1-10)]) I; lpd-6(sam151[LPD-6delRGG::M3])I;nucl-1(sam147[NUCL-1(deletion)::M3]) IV. **DUP265** glh-1(sam129[glh-1::T2A::sGFP2(1-10)]) I; garr-1(sam149[GARR-1::M3])IV;fib-1(sam144[fib-1::wrmScarlett])V. Sequence files for CRISPR-generated alleles are stored on figshare (see Data Availability Statement). All strains generated for this study are available upon request.

### CRISPR strain construction

CRISPR/Cas9 genome editing was used to place a C-terminal split superfolder-GFP11 tag on endogenous NUCL-1 (DUP229), LPD-6

(DUP249), and GARR-1 (DUP250) in the DUP218 background (GLH-1::T2A::sGFP2(1-10)), which drives sGFP2(1-10) in the germline. Creation of the DUP218 *glh-1(sam129[glh-1::T2A::sGFP2(1-10)]) I* allele was previously described[36]. RG/RGG domains were deleted from the sGFP11-expressing strains to create NUCL-1ΔRGG::sGFP11 (DUP239), LPD-6ΔRGG::sGFP11 (DUP252), GARR-1ΔNRGG::sGFP11 (DUP253), and GARR-1ΔCRGG::sGFP11 (DUP260). NUCL-1::sGFP11;FIB-1ΔRGG (DUP244), ΔNUCL-1::sGFP11 (DUP248) and NUCL-1::sGFP11;FIB-1::wrmScarlet (DUP243) were made by injecting CRISPR constructs into NUCL-1::sGFP11 (DUP229). NUCL-1ΔRGG::sGFP11;FIB-1ΔRGG (DUP245) and NUCL-1(FIB-1RGG)::sGFP11 (DUP257) was made by injecting CRISPR constructs into NUCL-1ΔRGG::sGFP11 (DUP239). CRISPR techniques for efficient genome editing in *C. elegans* were followed as described[61]. All CRISPR reagents (Cas9 (Cat# 1081058), trRNA (Cat# 1072532), crRNAs (2nmol), and ssDNA repair templates (Ultramer DNA oligo)) were ordered from Integrated DNA Technologies, Inc (San Diego, CA). Sequences for the guide RNA and repair templates are stored on figshare (https://figshare.com/articles/journal_contribution/Spaulding_et_al_CRISPR_Reagents_xlsx/17704127). All edits generated for this study were sequence verified and sequence files are stored on figshare (https://figshare.com/articles/dataset/CRISPR_Sequences/21231851). All strains generated for this study are available upon request.

## Survey of RG/RGG repeat-containing proteins in *C. elegans*
We defined the RG/RGG repeat domain as 3 or more instances of arginine-glycine (RG) followed by zero to any number of additional glycine residues (RG + ). Each instance of RG + could be separated by up to 15 amino acid residues of any type (RG + $_{(0-15)}$RG + $_{(0-15)}$RG + …). These criteria were converted into a Regular Expression (RegEX) sequence written as $RG + .\{0,15\}\{2,\}RG + $. The *C. elegans* proteome was downloaded in FASTA format from Ensembl BioMart on July 2$^{nd}$, 2020. The unix command "grep" tool was used with the RegEX sequence above to extract RG/RGG repeat-containing proteins from the BioMart file. 525 proteins satisfied these criteria.

*C. elegans* proteins that contained one of the top 50 longest RG/RGG sequences were loaded into STRING to determine gene ontology and SMART protein domain enrichments[62]. RG/RGG sequences from *C. elegans* proteins annotated to the Gene Ontology terms "nucleolus" and "p granule" and RG/RGG sequences from human nucleolar proteins were separately uploaded into the MEME tool from "The MEME Suite" to search for novel motifs.

## Brood size
Brood size was determined by plating a total of 11-24 individual L4 worms in at least two independent experiments at either 15 °C, 20 °C, or 25 °C. Hatched F1 progeny were counted for each worm.

## Embryo hatching
Embryo hatching was determined by picking 8 adult worms to new plates and allowing them to lay embryos for 3–4 h at 20 °C. Adults were then removed from the plates and embryos were marked, counted, and stored overnight at 20 °C. Unhatched embryos were counted again either 18 or 24 h later. This analysis was performed 5-12 times for each genotype.

## Fertility
For each strain, 20 L4 worms were picked to plates and allowed to develop into adults at either 15 °C, 20 °C, or 25 °C. The percentage of fertile or grotty (uterus filled with unfertilized oocytes and terminal embryos) compared to clean (germline atrophy with empty uterus) was scored on day 2-3 of adulthood. This analysis was performed at least 5 times for each genotype and temperature.

## Time to vulval development
Gravid adult worms were placed into drops of 1:1:1 water:bleach:4MNaOH on plates with no food source and placed at 20 °C overnight. The next day approximately 50-100 hatched L1 larvae were transferred to plates with an OP50 lawn. After 48 h of growth at 20 °C worms were observed through a dissecting microscope every hour and removed from the plate and recorded when full vulval development was reached. Full vulval development was reached when exterior vulval lips could be observed.

## Nucleolar imaging and analysis
L4 worms were plated at 20 °C the day prior to imaging. On the day of imaging live young adult worms were mounted on agarose pads in egg buffer (25 mM HEPES (Fisher, cat#BP310-1), 120 mM NaCl (Sigma, cat#S9888, 2 mM MgCl$_2$ (Sigma, cat#M9272, 50 mM KCl (Fisher, cat#S77375-1, and 5 or 10 mM levamisole(Thermofisher, cat# AC187870100) between slide and No.1.5 coverslip (Fisherbrand). Images were acquired using a point scanning confocal unit (LSM 980, Carl Zeiss Microscopy, Germany) on a Zeiss Axio Examiner Z1 upright microscope stand (ref: 409000-9752-000, Carl Zeiss Microscopy, Germany) equipped with either a Plan/Apochromat 10X/0.45 (ref:420640-9900-000, Carl Zeiss Microscopy, Germany) or Plan-Apochromat ×63/1.4 Oil (ref:420782-9900-799, Carl Zeiss Microscopy, Germany) objective.

sGFP and wrmScarlet fluorescence were excited with the 488 nm Diode (0.2–1.2% laser power) and the 561 nm DPSS laser (0.5% power), respectively. Fluorescence was collected with Airyscan2 with the following detection wavelengths: sGFP from 499 to 557 nm and wrmScarlet from 573–627 nm. Images were sequentially acquired in Super Resolution mode (SR) at zoom 10, with a line average of 1, a resolution of 292 × 292 pixels, 0.043 × 0.043 mm pixel size, a pixel time of 0.69 ms, in 16-bit, and in bidirectional mode. Z stacks taken with the ×10 objective were acquired using standard confocal mode. Z-stack images were collected with a step size of 0.170 mm with the Motorized Scanning Stage 130 × 85 PIEZO (Carl Zeiss Microscopy) mounted on Z-piezo stage insert WSB500 (Carl Zeiss Microscopy). Microscope was controlled using Zen Blue Software (Zen Pro 3.1), Airy scan images were processed using the "auto" mode and saved in CZI format. For coefficient of variation experiments with NUCL-1(FIBRGG), the gain was optimized to acquire quality images despite its comparatively low expression. Z stacks of 10–20 pachytene germ cell nucleoli were taken from each of 10–12 worms per genotype during 3–4 separate experiments.

To image DIC/NUCL-1, the 488 nm laser with 2.00% laser power was used. The DIC image was collected using the T-PMT detector (ref:000000–2183), and the GFP fluorescence was collected using the spectral detector with a detection range of 499–552 nm. Images were sequentially acquired using the Plan-Apochromat ×63/1.4 Oil (ref:420782-9900-799, Carl Zeiss Microscopy, Germany) objective, at zoom 10, with a resolution of 158 × 158 pixels, a pixel size of 0.085 × 0.085 μm, a pixel time of 53.43 μs,16-bit, in Bidirectional mode and controlled with Zen Blue software (Zen Pro 3.1).

3D reconstructions of Z stacks of 6–10 nucleoli from 9–10 worms of each genotype were created using the Imaris 9.7 software. The volume of whole nucleoli that fit within the Z stack was measured using the volume feature. Briefly, background subtraction parameters were set by first measuring the diameter of the largest sphere which fit into nucleoli. Automatic thresholding was used to first create 3D volumes of all objects in the image, and then visual inspection was used to narrow down volumes to only include nucleoli whose entire volume fit into the z stack. ImageJ was used to quantify the coefficient of variation in individual nucleoli. A single plane of the Z stack was chosen for each nucleolus that contained the maximum nucleolar area. A circle was

placed within individual nucleoli that covered its maximum area without including background space and the following macro code was used to calculate the coefficient of variation (standard-deviation divided by the mean fluorescence):

    getRawStatistics(N, mean, min, max, std);
    print(std / mean);

ImageJ was also used to quantify fluorescence intensity of individual nucleoli. A circle of fixed size was placed into individual nucleoli and fluorescence intensity was measured. The same circle was used to quantify background intensity of three spots adjacent to each nucleolus. The average intensity of these 3 background spots was subtracted from the intensity of the nucleolus. Except for NUCL-1(FIBRGG), the same images were used for quantification of nucleolar volume, CV, and fluorescence intensity. For NUCL-1(FIBRGG), separate images were taken for CV experiments with optimized gain settings to allow for sufficient visualization of protein homogeneity/dispersal.

ImageJ was used to create profile plots of single WT NUCL-1;FIB-1 and NUCL-1ΔRGG;FIB-1-labeled nucleoli to show localization patterns. Dual-channel images were split into 2 images and the frames were synchronized. Brightness was auto-scaled for both GFP and wrmScarlet images. A straight line was drawn across the middle of a single nucleolus. A plot profile was created for each image and the data (distance and gray values) were copied into Prism to create a single plot.

### NUCL-1/FIB-1 genetic interaction

The RG/RGG repeats of endogenous FIB-1 were deleted from both NUCL-1::sGFP11 and NUCL-1ΔRGG::sGFP11 strains in two different CRISPR experimental replicates. 17 F1 worms from successful WT NUCL-1 and 16 F1 worms from successful NUCL-1ΔRGG injections were genotyped and only heterozygous FIB-1RG/RGG deletions were recovered. 10–20 WT NUCL-1 worms from each of the F2 through F5 generations and 25 NUCL-1ΔRGG worms from the F2 generation were genotyped to confirm the difference in maintaining heterozygosity on each background.

### Worm crosses

Males were generated by plating 10 L4 hermaphrodites each on 10 plates and incubating at 30 °C for 6 h. Plates were then shifted to 20 °C and males were picked 3 days later. DUP248 males were crossed into DUP252 hermaphrodites. 4 F1 worms were picked from plates with 50% males (indicating successful mating). 12 F2 worms were picked from each F1 clone. F3 worms were genotyped for LPD-6ΔRGG, and worms that lacked the LPD-6 RGG domain were then genotyped for NUCL-1 deletion. DUP250 males were crossed into DUP243 hermaphrodites. 4 F1 worms with FIB-1::wrmScarlet expression were picked from plates with 50% males (indicating successful mating). 12 F2 worms with FIB-1::wrmScarlet expression were picked from each F1 clone. F3 worms were genotyped for the presence of GARR-1::M3 and the lack of NUCL-1::M3.

### Western blotting

For NUCL-1 strains, 300 worms of each genotype were picked directly into 2× Laemmli buffer (10% beta-mercaptoethanol (Fisher, cat#BP176-100), 4% SDS (Bio-Rad, cat#161-0203), 20% glycerol (Invitrogen, cat#15514-011), .004% bromphenol blue (Sigma, cat#B5525), 0.125 M Tris-Cl pH6.8 (Fisher, cat#BP153-500)). For GARR-1 and LPD-6 strains, 3–4 starved plates of worms were washed into an Eppendorf tube with water. Tubes were spun down for 1 minute at 2 K and 1:1 Laemmli buffer was added. All samples were then boiled for 5 min and spun at 12 K for 5 min at room temperature. The supernatant was loaded onto a mini-PROTEAN TGX stain-free gel (Bio-Rad, cat#4568084) and run at 200 V for 25 min in SDS running buffer (.2501 M Tris base (Fisher, cat#BP15201), 1.924 M glycine (Fisher, cat#G48-500), .0347 M SDS (Bio-Rad, cat#161-0203)). The contents of the gel were transferred using a trans-blot turbo transfer pack (Bio-Rad, cat#1704156) on the Bio-RadTrans-Blot Turbo Transfer System. The PVDF membrane was then blocked in 5% nonfat milk in TBST (20 mM Tris-HCl pH7.4, 150 mM NaCl, 0.1% Tween) at room temperature for 30 min. The membrane was incubated overnight at 4 °C with rabbit polyclonal anti-GFP11, 1:2000 (MyBioSource, product name:MBS8565385, lot#J14109) in 5% milk and then washed 3 × 10 min in TBST at room temperature. The membrane was then incubated in goat anti-rabbit IgG-HRP, 1:5000 (Bio-Rad, cat#170-6515) in 5% milk for 1 hour at room temperature and washed 3 × 10 min in TBST. Finally, the membrane was developed using Clarity Western ECL Substrate (Bio-Rad, cat#1705060 S) and imaged on a Syngene G:Box gel and blot imaging system. For images of uncropped blots see the Source Data file.

### Statistics and reproducibility

All imaging experiments investigating nucleolar accumulation and sub-nucleolar organization of WT and mutant RG/RGG proteins (Figs. 2b–h, 3e–g, 4b, b', f, i, j, 6c and Supplemental Figs. 3a, 4a, c) were performed on at least two independent occasions and similar results were always obtained. Western blots were performed on at least two independent occasions and similar results were obtained (Supplemental Fig. 3b–d). Plots of NUCL-1 and FIB-1 colocalization were made for at least three nucleoli, all with similar results (Supplemental Fig. 4b, d). PCR genotyping of ΔNUCL-1 worms was performed at least three independent times, with the same results (Supplemental Fig. 5b).

### Data availability

For CRISPR/Cas9 editing experiments, sequences for the guide RNA and repair templates are stored on figshare (https://figshare.com/articles/journal_contribution/Spaulding_et_al_CRISPR_Reagents_xlsx/17704127). All edits generated for this study were sequence verified and sequence files are stored on figshare (https://figshare.com/articles/dataset/CRISPR_Sequences/21231851). All strains generated for this study are available upon request. Source data for all imaging and in vivo experiments are provided with the paper. Source data are provided with this paper.

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

## Acknowledgements

We would like to acknowledge support from Chris Smith in the MDI Biological Laboratory (MDIBL) Sequencing Facility, Frederic Bonnet in the MDIBL Light Microscopy Facility, Joel Graber in the MDIBL Bioinformatics Core, and plate pouring services provided by the MDIBL Animal Resources Core (Cores and Facilities supported by the Maine INBRE NIH P20GM103423 and the MDIBL COBRE NIH P20GM104318). Funding: NSF REU DBI-1460495 (A.M.F), Maine INBRE fellowship NIH P20GM103423 (L.A.C.), NRSA NIH F32GM143851 (E.L.S.), NIH R01GM113933 (D.L.U.). Schematics created with BioRender.com.

## Author contributions

A.F performed the in silico RG/RGG search and MEME motif analysis. L.C. generated the NUCL-1 deletion worms. E.S. generated all additional worm strains and performed in vivo experiments, imaging studies, and data analysis. D.U. and E.S. provided conceptualization and experimental design and wrote and edited the manuscript with input from all authors.

## Competing interests

The authors declare no competing interests.
