## [Peer Review File · Nature Communications]

REVIEWER COMMENTS

Reviewer #1 (Remarks to the Author):

Review for Spaulding et al 2022

Summary

The nucleolus is an intranuclear organelle with a conserved function in ribosome biogenesis. Nucleoli are organized into distinct layers: the fibrillar center (FC) where nascent rRNA is transcribed, the dense fibrillar component (DFC) where rRNA is processed, and the outer granular component (GC) where ribosomes are assembled. The substructure of the nucleolus has been proposed to facilitate “assembly line” processing of rRNA and ribosome assembly by segregating factors that function sequentially to distinct inner, middle and outer layers (Feric et al 2016, Lafontaine et al 2021).

Spaulding et al. conducted a survey of RGG repeat-containing proteins in *C. elegans* and in the process identified several proteins that localize to nucleoli. The survey led them to NUCL-1, a previously uncharacterized protein they argue is the *C. elegans* homolog of nucleolin, a protein in the GC layer. They show that NUCL-1 lacking RGG domains still accumulates in nucleoli but no longer enriches in a distinct layered pattern. The distribution of Fibrillarin, an FC layer component, is also affected in the NUCL-1(RGG delete) mutants. Remarkably these mutants develop normally and have normal brood sizes, raising the possibility that nucleolar substructure is non-essential in *C. elegans* (but see below).

This study makes some very intriguing observations and establishes *C. elegans* as a strong model to relate nucleolar substructure to function using sophisticated genetic tools. The possibility that nucleolar substructure is dispensable for nucleolar function is very intriguing and goes against many assumptions in the field and will certainly be of interest to many. The data are very rich, however, they are poorly organized, and some critical controls are missing. I list below recommendations to improve the manuscript in order of importance.

Major comments

1. In Fig. 1 the authors state that NUCL-1 localizes to nucleoli but show no independent marker to confirm this. The authors need to use known bona fide nucleolar markers to define nucleoli and each layer (e.g. FIB-1 for the FC layer). The authors should also describe their tagging strategy (split GFP) in more detail in the text and justify that this approach is suitable for studying nucleolar layers. Are all nucleolar layers accessible to GFP1-10? Does GFP11 fused to a neutral carrier protein show uniform enrichment in all nucleolar layers?
2. What is the evidence that nucleoli in *C. elegans* have only two layers? In line 97 the authors describe nucleoli as having a 2-layered structure and use the distribution of the newly identified NUCL-1 as evidence. However, in Fig 1F, the distribution of NUCL-1 suggests three layers: an outer “GC” that contains NUCL-1, a middle layer devoid of NUCL-1, and a central core enriched for NUCL-1. The authors need to define the layers using bona fide markers.
3. In Fig. 2H, the authors show that NUCL-1(RGGdelete) no longer localizes in distinct layers and overlaps with FIB-1, which also appears disorganized. They conclude that the mutant lacks “nucleolar sub-compartmentalization”. This is a key result. However close examination of the merged panels in Fig. 2H suggests that, despite a more dispersed appearance, NUCL-1(RGGdelete) and FIB-1 are not homogeneously distributed and in fact may still be non-overlapping. This observation suggests that distinct GC/FC layers may be preserved in the mutant but at a finer scale. One possibility is that in the absence of the RGG domain, microscopic GC compartments still form but now fail to fuse to form a continuous macroscopic GC layer, and consequently FC compartments also remain separated. This is a very interesting possibility that could be addressed by higher resolution microscopy and/or by examining other cell types with larger nucleoli. If the required technology is not available, the authors should at least mention this hypothesis, since it challenges directly the main hypothesis of the paper: spatial organization is not required for nucleolar function.

4. The finding that NUCL-1(RGGdelete) mutants are viable and develop at a normal rate is very intriguing and would be even more so if the authors could demonstrate that this mutant not only disrupts nucleolar organization in the germline but also in the soma. At a minimum, they should address this by examining the distribution of FIB-1 in somatic cells in the NUCL-1(RGGdelete) line expressing tagged FIB-1::wrmScarlet.

5. The authors should reorganize the paper as follows:
Consider separating introduction, results, and conclusion
Organize results systematically:

1. Sequence analyses of RGG motif proteins – Build a figure with all the newly identified proteins, including full domain structure and corresponding human homologs
2. Definition of *C. elegans* nucleoli and nucleolar layers using bona fide markers.
3. Tagging and localization of newly identified proteins using double-labeling experiments with known markers for assignment to specific layers.
4. Deletion of the RGG repeats and effects on sub-nucleolar organization
5. Replacement of the RGG repeats and effects on sub-nucleolar organization
6. Effects of null and RGG mutants on development and brood size

6. In the beginning of the main text, the authors should consider a more thorough review of the “assembly line” model for ribosome biogenesis in nucleoli. Emphasis on the model in which nucleolar compartmentalization is critical for function would increase the impact of the findings in this present study. The authors might add a cartoon or schematic to Fig 1 to clearly illustrate this model for unfamiliar audiences. Additionally, the authors should discuss what is known about the functions of nucleolin and *gar1* in other organisms: has either protein been implicated as an important scaffold for nucleolar compartmentalization?

Additional comments:

7. The authors use a schematic in Figure 3B to illustrate the synthetic lethality of NUCL-1 and FIB-1 RGG mutants. Here the authors should include the sample size of each experiment and ratios of genotypes for the F1 and F2 generations.

8. The authors generate a NUCL-1/FIB-1RGG chimera to test whether the RGG domain of FIB-1 can compensate for loss of the NUCL-1 RGG domain. The authors find that the FIB-1 RGG domain does not functionally replace the NUCL-1 RGG domain (Fig 3E and F). However, by western blot in Figure S2F the NUCL-1/FIB-1RGG chimera appears to be barely expressed, therefore the lack of rescue is complicated to interpret.

9. In Fig S3, the authors find that a third RGG-domain protein, LPD-6, localizes to nucleoli but report that it might not be enriched in a distinct subcompartment. However, LPD-6 does not appear uniform in the images shown. What is known about the localization of LPD-6 in other systems? Could the authors use a more sensitive method to quantify LPD-6 homogeneity with free GFP as a negative control?

Reviewer #2 (Remarks to the Author):

This manuscript from the Updike lab takes a bioinformatic approach to identify the *C. elegans* homolog of Nucleolin and includes a functional analysis of the RG/RGG repeats in Nucleolin and a second nucleolar protein GARR-1. While the identification of a nematode homolog of nucleolin is notable, as it was previously thought to be absent, it is the functional analysis of the RGG domains that is particularly significant. This study is timely as an emerging paradigm of cellular organization of RNA and RNA binding proteins is the formation of membraneless organelles, such as the nucleolus. Spaulding et al. show that the RG/RGG repeats are dispensable for NUCL-1 and GARR-1 to accumulate in nucleoli but are required for phase separation of the proteins. Moreover, a physiological function is demonstrated for the RG/RGG domain of GARR-1 (but not NUCL-1) in fertility and development. The

de-coupling (to at least some extent) of sub-nucleolar phase separation and nucleolar function is significant and will impact future analyses of ribosome biogenesis.

This manuscript is well-written and is accessible, flows logically, includes thorough methodology with appropriate validation and analyses, and will be of high interest to researchers in the fields of RNA biology and cell biology, and those studying the role of nucleoli in neurodegenerative disease and cancer. A few minor modifications will strengthen and clarify the manuscript for readers.

Comments

In Fig. 1, NUCL-1 is detected in nucleoli throughout the adult germline. Is it restricted to the germline, or also detected in somatic cells?

In Fig. 1F, the super-resolution images are beautiful. For the uninitiated, could a cartoon be added to better explain the three phase-separated layers of nucleoli in higher eukaryotes (FC, GC, and DFC)?

Lines 131 and 140 seem contradictory. First RGG repeats are required for mouse NCL nucleolar accumulation, but then the acidic domain or RRM is said to be key?

Line 151. As described, it sounds like a high coefficient of variation is associated with homogeneity, but it's lower in NUCL-1 Δ RGG worms, so this could be more clearly explained in the text, especially because this measure is used in multiple experiments.

Fig. 2E. Is the pink arrow in the correct position? Should be indicating areas of depletion, but seems to be one of the two peaks of enrichment?

Line 193-196. The significantly decreased expression of NUCL-1 (FIB-1RGG) is shown not only in extended Fig. 2I, but also in the Western blot (extended Fig. 2F). An additional sentence speculating on this seemingly unexpected and large decrease in protein level seems appropriate.

Minor comments

At line 71, the authors refer to K07H8.10(NUCL-1) as highly abundant, but the basis of this descriptor could be clarified.

Line 114. Adding the rationale or context for examining embryonic viability after 5 and 40 generations, and in line 238, brood sizes after 80-90 generations would be valuable to readers.

Line 125. Is it possible deletion of NUCL-1 in the nematode is viable because of redundancy? With the bioinformatic analyses, it seems this possibility could be addressed immediately, and it comes up later in Fig. 3 with FIB-1, and Fig. 4 with GARR-1.

Line 136: Clarify that there is no difference in nucleolar volume of NUCL-1 Δ RGG worms (instead of stating expression). The analysis is more precise, and this is an important quantitation that could be moved to Fig. 2 (alongside 2B).

Fig. 2H. Add GC and FC labels to arrows ;the blue colored arrow is somewhat hard to distinguish from the white-colored arrow.

A few figure labels could be simplified to enhance clarity:

- Fig. 2B, 3D. Could the GLH-1 part of the strain be omitted from the figure label? It is not included in the legends.
- Enhanced fig. 2G,H,I. Could the Y axes of the graphs omit the 'background subtracted' detail of the method? If important, it could be included in the figure legend.

Reviewer #3 (Remarks to the Author):

Review of "RG/RGG repeats in the *C. elegans* homologs of Nucleolin and GAR1 contribute to subnucleolar phase separation"

Overview

RG/RGG repeats are common in intrinsically disordered proteins present in biomolecular condensates, but their molecular functions are poorly understood. Spaulding and colleagues report a genome-wide survey for *C. elegans* proteins with multiple RG/RGG repeats, find many proteins, sort them by organelle and identify variant RG/RGG motifs specific to two subcellular organelles, the nucleolus or P granules. Then, they identify NUCL-1 and GARR-1, worm homologs of the broadly conserved nucleolar proteins Nucleolin and GAR1. They next generate a *nucl-1* deletion mutant and find, surprisingly, that this null mutant is viable with only minor growth defects. Finally, they present a series of elegant experiments that rely on CRISPR-induced deletions of RG/RGG repeats in multiple genes followed by imaging to explore effects of the repeats on nucleolar accumulation and nucleolar compartmentalization. They find that RG/RGG repeats of NUCL-1 and GARR-1 are essential for compartmentalization within the nucleolus, but not for nucleolar localization. This discovery provides an exciting advance in understanding the role of RGG repeats in subcellular compartmentalization and will be of broad interest.

Critique

I have no major criticisms of this work. The data in the manuscript strongly support its conclusions, figures are clear and well annotated, and the manuscript is well written. Most importantly, the paper presents new and fascinating insights into the control of nucleolar compartmentalization. A major strength is that all experiments probe nucleolar structure in an intact organism and that all structure-function analyses are done with precisely engineered CRISPR-induced mutants in endogenous genes. This paper will clearly make an important contribution if published as submitted. However, I offer a few suggestions for the authors to consider.

1. The identification of organelle-specific variant RGG motifs is a nice result and would be best presented without having to access supp data for half of it. Could Figures 1C and extended data 1D be placed next to each other in the same figure. Indeed, why not include all bioinformatic analyses in figure 1 (moving extended figures 1B-D to figure 1), and use Figure 2 to report discovery of NUCL-1 and its null phenotype?
2. The statement that K07H8.10 accumulates in the nucleolus (lines 89-90) cites Figure 1E, but this image does not include any marker of the nucleolus or even the nucleus. The round blobs in Figure 1E are not convincingly nucleoli. A simple solution would be to cite Figures 1E and 1F, since staining in 1F is distinctly nucleolar. A more labor intensive solution would be to add a marker of the nucleolus or at least the nucleus to Figure 1E. Once established, this would not be necessary in subsequent figures.
3. Deletion of the FIB-1 RGG repeats was generated to test a genetic interaction with the NUCL-1 RGG deletion mutant. However, that deletion could also be used to ask what effects the FIB-1 RGG repeats have on nucleolar accumulation of FIB-1 and on nucleolar compartmentalization. This paper is already chock full of interesting results, and I am loathe to ask for more, but RGG function in nucleolar accumulation and compartmentalization is central to the paper and the reagents are in hand to answer it for FIB-1. Omission of this experiment left me wondering why the question was not addressed? Also, might NUCL-1 compartmentalization depend on the FIB-1 RGG repeats, much like FIB-1 compartmentalization depends on the NUCL-1 RGG repeats?
4. A final figure would be nice -- with a table that pulls together the data for all four proteins you have assayed plus a model for how the various RGG domains affect nucleolar compartmentalization.

Reviewer #4 (Remarks to the Author):

The manuscript by Spaulding et al. (Dustin Updike's lab) surveyed all RGG/RG proteins in *C. elegans* (Extended Data Figure 1), and in so doing they identified a Nucleolin orthologue in *C. elegans* for the first time. The authors then examined the properties of RGG/RG motifs in Nucleolin, Fibrillarin, the GAR1 homologue they call GARR1, and in LPD-6 for their role in sub-nucleolar phase separation. Interesting, deleting the RGG/RG domain from Nucleolin did not prevent its localization within nucleoli as RGG deletion as it does in higher organisms, but the deletion disrupted sub-nucleolar structure in *C. elegans*. The authors argue that the low complexity RGG/RG domains are intrinsically disordered and function in nucleolar phase separation to organize the *C. elegans* nucleolus.

This is a timely study, as much attention now focuses on Liquid-Liquid Phase Separation (LLSP) in the formation of nucleolar sub-compartments (Lafontaine et al., 2021; <https://www.nature.com/articles/s41580-020-0272-6>). This manuscript by Spaulding et al. should generate much discussion in this field. The manuscript is very well written and the figures are excellent.

The manuscript should be accepted for publication after minor revisions/clarifications as listed here:

1) The overall structure of nucleoli in *C. elegans* is intriguing. As a "primitive" organism, the nucleoli contain Fibrillar Centers (FCs) and Granular Components (GCs), but they lack the more recently evolved Dense Fibrillar Regions (DFCs). I'm interested in how these nucleoli appear by simple phase contrast microscopy as they compare to nucleoli in vertebrates. This is particularly relevant because mutating NUCL-1, a GC protein, adversely affects the location of Fibrillarin, an FC protein.

2) The sentence in lines 106-107, "NCL [Nucleolin] also interacts with dipeptide repeat proteins...and is a modifier of dipeptide toxicity in *Drosophila*" is confusing because a close Nucleolin orthologue has not been identified in *Drosophila*. The Extended Data Figure 2 in this current manuscript correctly describes the status of NCL in *Drosophila* as "unclear". While reference #23 in line 107 (Lee et al. 2016) describes nucleolin in *Drosophila*, these authors expressed human nucleolin in *Drosophila* cells, and the NCL orthologue they cross-referenced in their Figure 2 is Cysteine string protein (Csp) which encodes a synaptic vesicle-associated co-chaperone of Hsc70 that is vital for presynaptic proteostasis and maintenance of synaptic function. In other words, Csp has nothing to do with nucleoli. I have no idea what the authors of reference 23 are talking about.

3) Deleting NUCL-1 did not cause lethality, but deletion caused delayed development. Do you think another GC protein (e.g., LPD-6) could partially substitute for NUCL-1 in function?

4) Deleting the RGG/RG domain of NUCL-1 did not affect NUCL-1 Δ RGG's localization to nucleoli, but it did disrupt nucleolar morphology, causing Fibrillarin to redistribute partially from the FCs. In Line 158, you say loss of NUCL's RGG/RG domain "may not impact nucleolar function." This was based on the observation that there was no apparent change in nucleolar volume. But you still haven't tested nucleolar function in the absence of NUCL-1's RGG/RG domain. I would suggest deleting the phrase in lines 157-159 ("suggesting that loss...").

5) Did GARR-1 redistribute from the FC's with the loss of the NUCL-1 RGG/RG domain?

6) Just curious, have you looked for methylation (asymmetrical di-methylation) on these RGG/RG motifs in the *C. elegans* proteins?

7) The genetic interactions between NUCL-1 Δ RGG and FIB Δ RGG in Figure 3 suggest a partial redundancy of their respective RGG/RG domains. Are you saying the RGG/RG domain of NUCL-1 can functionally substitute for the RGG/RG domain in Fibrillarin even when the two proteins occupy different sub-regions of the nucleoli? The concern I have is: Fibrillarin is an integral component of C/D box snoRNP complexes, and its RGG/RG domain may be important for assembly and function of these snoRNPs, and not so much for overall nucleolar structure. Perhaps GARR1's two RGG/RG domains are responsible for FC structure instead of Fibrillarin's single RGG/RG domain.

Have you looked at nucleolar structure when just FIB Δ RGG is expressed in an otherwise WT NUCL-1 background? Admittedly, interpreting results from this experiment may be difficult as a loss in C/D box snoRNPs may be catastrophic for nucleolar structure and function.

8) Do you think a critical threshold concentration of RGG/RG's is required for nucleolar assembly (LLPS), and it doesn't matter which protein (NUCL-1, GARR1, or LPD-6) provides the RGG/RG motifs for LLPS? This might explain why deleting the RGG's from either NUCL-1 or LPD-6 had no drastic effect because enough were still provided by the other proteins present.

REVIEWER COMMENTS

Reviewer #1 (Remarks to the Author):

Review for Spaulding et al 2022

Summary

The nucleolus is an intranuclear organelle with a conserved function in ribosome biogenesis. Nucleoli are organized into distinct layers: the fibrillar center (FC) where nascent rRNA is transcribed, the dense fibrillar component (DFC) where rRNA is processed, and the outer granular component (GC) where ribosomes are assembled. The substructure of the nucleolus has been proposed to facilitate “assembly line” processing of rRNA and ribosome assembly by segregating factors that function sequentially to distinct inner, middle and outer layers (Feric et al 2016, Lafontaine et al 2021).

Spaulding et al. conducted a survey of RGG repeat-containing proteins in *C. elegans* and in the process identified several proteins that localize to nucleoli. The survey led them to NUCL-1, a previously uncharacterized protein they argue is the *C. elegans* homolog of nucleolin, a protein in the GC layer. They show that NUCL-1 lacking RGG domains still accumulates in nucleoli but no longer enriches in a distinct layered pattern. The distribution of Fibrillarin, an FC layer component, is also affected in the NUCL-1(RGG delete) mutants. Remarkably these mutants develop normally and have normal brood sizes, raising the possibility that nucleolar substructure is non-essential in *C. elegans* (but see below).

This study makes some very intriguing observations and establishes *C. elegans* as a strong model to relate nucleolar substructure to function using sophisticated genetic tools. The possibility that nucleolar substructure is dispensable for nucleolar function is very intriguing and goes against many assumptions in the field and will certainly be of interest to many. The data are very rich, however, they are poorly organized, and some critical controls are missing. I list below recommendations to improve the manuscript in order of importance.

Major comments

1. In Fig. 1 the authors state that NUCL-1 localizes to nucleoli but show no independent marker to confirm this. The authors need to use known bona fide nucleolar markers to define nucleoli and each layer (e.g. FIB-1 for the FC layer). The authors should also describe their tagging strategy (split GFP) in more detail in the text and justify that this approach is suitable for studying nucleolar layers. Are all nucleolar layers accessible to GFP1-10? Does GFP11 fused to a neutral carrier protein show uniform enrichment in all nucleolar layers?

We thank the reviewer for these suggestions. DIC/NUCL-1 imaging of oocyte and pachytene nucleoli has now been included in Figure 2c to clearly show NUCL-1 localization in nucleoli. No bona fide nucleolar layer markers exist in *C. elegans*. To our knowledge, we are the first to image endogenous nucleolar proteins with sufficiently high resolution to visualize sub-nucleolar layers. In addition, *C. elegans* does not have a homolog for Nucleophosmin, a protein commonly used as a GC marker in other animal models. However, to better define the proposed FC layer we have crossed the tagged GARR-1 line with the tagged FIB-1 line to clearly show the two proteins occupy the same nucleolar sub-compartment (Figure 2g).

The split GFP tagging strategy is now explained in more detail in the text (lines 128-136), and a schematic has been added to Figure 2a. To determine if all sub-nucleolar layers are equally accessible to GFP, we have performed super-resolution imaging of germ cells in a strain that expresses free GFP in the germline (GLH-1::T2A::sGFP1-11). GFP shows uniform enrichment throughout the nucleolus, as shown in an oocyte nucleolus in Supplementary Fig. 3a. We have also calculated the free GFP coefficient of variation in pachytene nucleoli and observe no evidence of organization into nucleolar layers (Fig. 5c,d).

2. What is the evidence that nucleoli in *C. elegans* have only two layers? In line 97 the authors describe nucleoli as having a 2-layered structure and use the distribution of the newly identified NUCL-1 as evidence. However, in Fig 1F, the distribution of NUCL-1 suggests three layers: an outer “GC” that contains NUCL-1, a middle layer devoid of NUCL-1, and a central core enriched for NUCL-1. The authors need to define the layers using bona fide markers.

The DIC/NUCL-1 imaging suggested by reviewer 1 now allows us to better interpret NUCL-1 super-resolution images. In the image mentioned by the reviewer (now Fig 2d), we show that the central “holes” are nucleolar vacuoles, rather than a 3rd nucleolar layer (see NUCL-1/DIC imaging in figure 2c). To better visualize the 2-layers marked by NUCL-1 and FIB-1 in Figure 2e, we have added videos of 3-dimensional reconstruction and volume rendering of a single pachytene nucleolus (Supplemental Videos 1 and 2). We believe these videos more clearly demonstrate NUCL-1 encapsulation of FIB-1.

3. In Fig. 2H, the authors show that NUCL-1(RGGdelete) no longer localizes in distinct layers and overlaps with FIB-1, which also appears disorganized. They conclude that the mutant lacks “nucleolar sub-compartmentalization”. This is a key result. However close examination of the merged panels in Fig. 2H suggests that, despite a more dispersed appearance, NUCL-1(RGGdelete) and FIB-1 are not homogeneously distributed and in fact may still be non-overlapping. This observation suggest that distinct GC/FC layers may be preserved in the mutant but at a finer scale. One possibility is that in the absence of the RGG domain, microscopic GC compartments still form but now fail to fuse to form a continuous macroscopic GC layer, and consequently FC compartments also remain separated. This is a very interesting possibility that could be addressed by higher resolution microscopy and/or by examining other cell types with larger nucleoli. If the required technology is not available, the authors should at least mention this hypothesis, since it challenges directly the main hypothesis of the paper: spatial organization is not required for nucleolar function.

We thank the reviewer for this valuable observation. Indeed, when observing NUCL-1 Δ RGG nucleoli, NUCL-1 and FIB-1 still show significant areas of non-overlapping expression, but on a finer scale. We have provided profile plots of NUCL-1 and FIB-1 fluorescence in individual WT and NUCL-1 Δ RGG nucleoli to demonstrate this point (Supplemental Figure 4). We have also added this point to the discussion (lines 378-381) and added accompanying videos through a Z stack of a single NUCL-1 Δ RGG nucleolus to demonstrate this point (Supplemental Videos 3 and 4). Although we agree that some micro-organization may still be occurring, our data still supports the possibility that canonical, large-scale organization may not be required for nucleolar function.

4. The finding that NUCL-1(RGGdelete) mutants are viable and develop at a normal rate is very intriguing and would be even more so if the authors could demonstrate that this mutant not only disrupts nucleolar organization in the germline but also in the soma. At a minimum, they should address this by examining the distribution of FIB-1 in somatic cells in the NUCL-1(RGGdelete) line expressing tagged FIB-1::wrmScarlet.

Thank you for this suggestion. Although creating a somatic NUCL-1 reporter line is out of the scope of this initial study, we have evaluated FIB-1 organization in NUCL-1 Δ RGG somatic cells. We have added these data to Figure 4i-l. We do find that NUCL-1 Δ RGG hypodermal nucleoli show impaired FIB-1 sub-nucleolar organization.

5. The authors should reorganize the paper as follows:

Consider separating introduction, results, and conclusion Organize results systematically:

1. Sequence analyses of RGG motif proteins – Build a figure with all the newly identified proteins, including full domain structure and corresponding human homologs
2. Definition of *C. elegans* nucleoli and nucleolar layers using bona fide markers.
3. Tagging and localization of newly identified proteins using double-labeling experiments with known markers for assignment to specific layers.
4. Deletion of the RGG repeats and effects on sub-nucleolar organization
5. Replacement of the RGG repeats and effects on sub-nucleolar organization
6. Effects of null and RGG mutants on development and brood size

We thank the reviewer for this suggestion and have reorganized the paper. We have not built a full additional figure with all newly identified RG/RGG proteins and their homologs, because for many, a human homolog is not clearly identifiable. However, we did add a panel to Figure 1 (Figure 1d) that includes the full domain structure of all nucleolar and P-granule RG/RGG proteins, as well as their human homologs.

6. In the beginning of the main text, the authors should consider a more thorough review of the “assembly line” model for ribosome biogenesis in nucleoli. Emphasis on the model in which nucleolar compartmentalization is critical for function would increase the impact of the findings in this present study. The authors might add a cartoon or schematic to Fig 1 to clearly illustrate this model for unfamiliar audiences. Additionally, the authors should discuss what is known about the functions of nucleolin and gar1 in other organisms: has either protein been implicated as an important scaffold for nucleolar compartmentalization?

We have included a schematic of nucleolar organization as it relates to the steps of ribosome biogenesis in Figure 2i and included a more thorough discussion in lines 186-192. To our knowledge, neither NCL nor GAR1 (or their RG/RGG domains) have been implicated in nucleolar compartmentalization.

Additional comments:

7. The authors use a schematic in Figure 3B to illustrate the synthetic lethality of NUCL-1 and FIB-1 RGG mutants. Here the authors should include the sample size of each experiment and ratios of genotypes for the F1 and F2 generations.

We have added the requested information to the figure (now Figure 6a).

8. The authors generate a NUCL-1/FIB-1RGG chimera to test whether the RGG domain of FIB-1 can compensate for loss of the NUCL-1 RGG domain. The authors find that the FIB-1 RGG domain does not functionally replace the NUCL-1 RGG domain (Fig 3E and F). However, by western blot in Figure S2F the NUCL-1/FIB-1RGG chimera appears to be barely expressed, therefore the lack of rescue is complicated to interpret.

This point is now more thoroughly discussed in lines 293-299.

9. In Fig S3, the authors find that a third RGG-domain protein, LPD-6, localizes to nucleoli but report that it might not be enriched in a distinct subcompartment. However, LPD-6 does not appear uniform in the images shown. What is known about the localization of LPD-6 in other systems? Could the authors use a more sensitive method to quantify LPD-6 homogeneity with free GFP as a negative control?

To our knowledge, the sub-nucleolar localization of LPD-6 homologs (*SSF1* in yeast and *PPAN* in humans) is unknown. Although we do not have a more sensitive method to quantify LPD-6 distribution within nucleoli, we agree with the author that LPD-6 does not appear uniform and have added free GFP as a negative control to the analysis of LPD-6 organization. Indeed, as shown in Figure 5c and d, LPD-6 has a significantly higher CV than free GFP, and we include in our observation that the localization of LPD-6 extends to the boundary of nucleoli, much like *NUCL-1* in the GC.

Reviewer #2 (Remarks to the Author):

This manuscript from the Updike lab takes a bioinformatic approach to identify the *C. elegans* homolog of Nucleolin and includes a functional analysis of the RG/RGG repeats in Nucleolin and a second nucleolar protein GARR-1. While the identification of a nematode homolog of nucleolin is notable, as it was previously thought to be absent, it is the functional analysis of the RGG domains that is particularly significant. This study is timely as an emerging paradigm of cellular organization of RNA and RNA binding proteins is the formation of membraneless organelles, such as the nucleolus. Spaulding et al. show that the RG/RGG repeats are dispensable for *NUCL-1* and *GARR-1* to accumulate in nucleoli but are required for phase separation of the proteins. Moreover, a physiological function is demonstrated for the RG/RGG domain of *GARR-1* (but not *NUCL-1*) in fertility and development. The de-coupling (to at least some extent) of sub-nucleolar phase separation and nucleolar function is significant and will impact future analyses of ribosome biogenesis.

This manuscript is well-written and is accessible, flows logically, includes thorough methodology with appropriate validation and analyses, and will be of high interest to researchers in the fields of RNA biology and cell biology, and those studying the role of nucleoli in neurodegenerative disease and cancer. A few minor modifications will strengthen and clarify the manuscript for readers.

Comments

In Fig. 1, *NUCL-1* is detected in nucleoli throughout the adult germline. Is it restricted to the germline, or also detected in somatic cells?

Our splitGFP system only allows for germline visualization of *NUCL-1*. However, our edits are on endogenous genes (so the modifications extend to all cells) and we anticipate that *NUCL-1* is expressed in all cell types. Our future plans include the creation of somatic *NUCL-1* reporter lines.

In Fig. 1F, the super-resolution images are beautiful. For the uninitiated, could a cartoon be added to better explain the three phase-separated layers of nucleoli in higher eukaryotes (FC, GC, and DFC)?

Thank you for this suggestion. We have added schematics of nucleolar layers in amniotes (higher eukaryotes), anamniotes (e.g., yeast), and *C. elegans* in Figure 2i-k.

Lines 131 and 140 seem contradictory. First RGG repeats are required for mouse NCL nucleolar accumulation, but then the acidic domain or RRM's are said to be key?

The reviewer is right that these lines seemed contradictory as written. We have edited the text to clarify (lines 68-70). Doron-Mandel, et al. found that in adult mouse dorsal root ganglia the NCL RGG domain is required for *complete* nucleolar accumulation of the protein. However, some of the RGG-deleted NCL still localizes to nucleoli. Because of this result, Doron-Mandel, et al. suggest that either the acidic or RNA binding domains of NCL may have the dominant role in nucleolar LLPS of the protein.

Line 151. As described, it sounds like a high coefficient of variation is associated with homogeneity, but it's lower in NUCL-1 Δ RGG worms, so this could be more clearly explained in the text, especially because this measure is used in multiple experiments.

Thank you for this suggestion, we have added a clearer description of what high and low coefficient of variation means in relation to nucleolar organization and our results in lines 225-230.

Fig. 2E. Is the pink arrow in the correct position? Should be indicating areas of depletion, but seems to be one of the two peaks of enrichment?

The pink arrow is in the correct position, but the reviewer is correct in pointing out this confusion. We have added x and y axes labels to the histograms (now Figure 4c) to clarify. The peak under the pink arrow corresponds to the pixels of low intensity, while the peak under the blue arrow corresponds to pixels of high intensity.

Line 193-196. The significantly decreased expression of NUCL-1 (FIB-1RGG) is shown not only in extended Fig. 2I, but also in the Western blot (extended Fig. 2F). An additional sentence speculating on this seemingly unexpected and large decrease in protein level seems appropriate.

We have added a discussion around this point in lines 293-299.

Minor comments

At line 71, the authors refer to K07H8.10(NUCL-1) as highly abundant, but the basis of this descriptor could be clarified.

Thank you for pointing this out. The characterization of K07H8.10 as "highly abundant" is based on RNA sequencing from dissected germlines in the now cited paper, Campbell and Updike, 2015. We have clarified that "highly abundant" refers to transcript abundance, not protein abundance (lines 112-114).

Line 114. Adding the rationale or context for examining embryonic viability after 5 and 40 generations, and in line 238, brood sizes after 80-90 generations would be valuable to readers.

Thank you for this suggestion. We have added text and cited a review article to explain the rationale for these experiments (lines 329-332).

Line 125. Is it possible deletion of NUCL-1 in the nematode is viable because of redundancy? With the bioinformatic analyses, it seems this possibility could be addressed immediately, and it comes up later in

Fig. 3 with FIB-1, and Fig. 4 with GARR-1.

We find no evidence of NUCL-1 paralogs in our bioinformatic analysis, but NUCL-1 may share redundancy with other nucleolar RG/RGG proteins, particularly those that localize to the GC. With this possibility in mind, we have crossed the Δ NUCL-1 strain with the LPD-6 Δ RGG strain and assayed fertility (Figure 7J). While Δ NUCL-1 and LPD-6 Δ RGG worms show no defect in brood size compared to WT, Δ NUCL-1;LPD-6 Δ RGG worms have decreased brood size. This result indicates LPD-6 may partially compensate for the loss of NUCL-1.

As the reviewer mentioned, we also demonstrate partial redundancy of the NUCL-1 and FIB-1 RG/RGG domains with genetic experiments in Figure 6. In light of these results, full deletion of NUCL-1 may also be viable because of compensation by the other FC protein, GARR-1. We have not genetically tested the potential redundancy of NUCL-1 and GARR-1 RGG domains because both genes are on the same chromosome, making crosses difficult.

The topic of NUCL-1 redundancy with other nucleolar RG/RGG proteins has also been added to the discussion (lines 391-402)

Line 136: Clarify that there is no difference in nucleolar volume of NUCL-1 Δ RGG worms (instead of stating expression). The analysis is more precise, and this is an important quantitation that could be moved to Fig. 2 (alongside 2B).

We have now clearly stated that nucleolar volume is not changed in NUCL-1 Δ RGG worms, and the data is in Figure 4e and discussed in lines 230-235.

Fig. 2H. Add GC and FC labels to arrows ;the blue colored arrow is somewhat hard to distinguish from the white-colored arrow.

Thank you for these suggestions, labels have been added and the blue arrow is now pink (now figure 4f).

A few figure labels could be simplified to enhance clarity:

- Fig. 2B, 3D. Could the GLH-1 part of the strain be omitted from the figure label? It is not included in the legends.

The GLH-1::sGFP1-10 portion of the strain was omitted from most figure labels.

- Enhanced fig. 2G,H,I. Could the Y axes of the graphs omit the 'background subtracted' detail of the method? If important, it could be included in the figure legend.

"Background subtracted" was omitted from the Y axes and moved to the figure legend.

Reviewer #3 (Remarks to the Author):

Review of "RG/RGG repeats in the C. elegans homologs of Nucleolin and GAR1 contribute to subnucleolar phase separation"

Overview

RG/RGG repeats are common in intrinsically disordered proteins present in biomolecular condensates, but their molecular functions are poorly understood. Spaulding and colleagues report a genome-wide survey for *C. elegans* proteins with multiple RG/RGG repeats, find many proteins, sort them by organelle and identify variant RG/RGG motifs specific to two subcellular organelles, the nucleolus or P granules. Then, they identify NUCL-1 and GARR-1, worm homologs of the broadly conserved nucleolar proteins Nucleolin and GAR1. They next generate a *nucl-1* deletion mutant and find, surprisingly, that this null mutant is viable with only minor growth defects. Finally, they present a series of elegant experiments that rely on CRISPR-induced deletions of RG/RGG repeats in multiple genes followed by imaging to explore effects of the repeats on nucleolar accumulation and nucleolar compartmentalization. They find that RG/RGG repeats of NUCL-1 and GARR-1 are essential for compartmentalization within the nucleolus, but not for nucleolar localization. This discovery provides an exciting advance in understanding the role of RGG repeats in subcellular compartmentalization and will be of broad interest.

Critique

I have no major criticisms of this work. The data in the manuscript strongly support its conclusions, figures are clear and well annotated, and the manuscript is well written. Most importantly, the paper presents new and fascinating insights into the control of nucleolar compartmentalization. A major strength is that all experiments probe nucleolar structure in an intact organism and that all structure-function analyses are done with precisely engineered CRISPR-induced mutants in endogenous genes. This paper will clearly make an important contribution if published as submitted. However, I offer a few suggestions for the authors to consider.

1. The identification of organelle-specific variant RGG motifs is a nice result and would be best presented without having to access supp data for half of it. Could Figures 1C and extended data 1D be placed next to each other in the same figure. Indeed, why not include all bioinformatic analyses in figure 1 (moving extended figures 1B-D to figure 1), and use Figure 2 to report discovery of NUCL-1 and its null phenotype?

Thank you for these suggestions. Most bioinformatics data have been moved to Figure 1, including all MEME motif analysis.

2. The statement that K07H8.10 accumulates in the nucleolus (lines 89-90) cites Figure 1E, but this image does not include any marker of the nucleolus or even the nucleus. The round blobs in Figure 1E are not convincingly nucleoli. A simple solution would be to cite Figures 1E and 1F, since staining in 1F is distinctly nucleolar. A more labor intensive solution would be to add a marker of the nucleolus or at least the nucleus to Figure 1E. Once established, this would not be necessary in subsequent figures.

We have now included DIC/NUCL-1 imaging in Figure 2C to clearly show NUCL-1 nucleolar localization.

3. Deletion of the FIB-1 RGG repeats was generated to test a genetic interaction with the NUCL-1 RGG deletion mutant. However, that deletion could also be used to ask what effects the FIB-1 RGG repeats have on nucleolar accumulation of FIB-1 and on nucleolar compartmentalization. This paper is already chock full of interesting results, and I am loathe to ask for more, but RGG function in nucleolar accumulation and compartmentalization is central to the paper and the reagents are in hand to answer it for FIB-1. Omission of this experiment left me wondering why the question was not addressed? Also, might NUCL-1 compartmentalization depend on the FIB-1 RGG repeats, much like FIB-1 compartmentalization depends on the NUCL-1 RGG repeats?

Our aim when creating the FIB-1 Δ RGG worms was to perform the exact experiments described by the reviewer. However, homozygous FIB-1 Δ RGG worms are not viable, so we are unable to answer these important questions.

4. A final figure would be nice -- with a table that pulls together the data for all four proteins you have assayed plus a model for how the various RGG domains affect nucleolar compartmentalization.

Thank you for this suggestion. We have added Figure 8 that includes a chart summarizing RG/RGG data and a model of how RGG domains may affect nucleolar compartmentalization.

Reviewer #4 (Remarks to the Author):

The manuscript by Spaulding et al. (Dustin Updike's lab) surveyed all RGG/RG proteins in *C. elegans* (Extended Data Figure 1), and in so doing they identified a Nucleolin orthologue in *C. elegans* for the first time. The authors then examined the properties of RGG/RG motifs in Nucleolin, Fibrillarin, the GAR1 homologue they call GARR1, and in LPD-6 for their role in sub-nucleolar phase separation. Interesting, deleting the RGG/RG domain from Nucleolin did not prevent its localization within nucleoli as RGG deletion as it does in higher organisms, but the deletion disrupted sub-nucleolar structure in *C. elegans*. The authors argue that the low complexity RGG/RG domains are intrinsically disordered and function in nucleolar phase separation to organize the *C. elegans* nucleolus.

This is a timely study, as much attention now focuses on Liquid-Liquid Phase Separation (LLSP) in the formation of nucleolar sub-compartments (Lafontaine et al., 2021; <https://www.nature.com/articles/s41580-020-0272-6>). This manuscript by Spaulding et al. should generate much discussion in this field. The manuscript is very well written and the figures are excellent.

The manuscript should be accepted for publication after minor revisions/clarifications as listed here:

1) The overall structure of nucleoli in *C. elegans* is intriguing. As a "primitive" organism, the nucleoli contain Fibrillar Centers (FCs) and Granular Components (GCs), but they lack the more recently evolved Dense Fibrillar Regions (DFCs). I'm interested in how these nucleoli appear by simple phase contrast microscopy as they compare to nucleoli in vertebrates. This is particularly relevant because mutating NUCL-1, a GC protein, adversely affects the location of Fibrillarin, an FC protein.

We have now added DIC/NUCL-1 imaging in Figure 2c.

2) The sentence in lines 106-107, "NCL [Nucleolin] also interacts with dipeptide repeat proteins...and is a modifier of dipeptide toxicity in *Drosophila*" is confusing because a close Nucleolin orthologue has not been identified in *Drosophila*. The Extended Data Figure 2 in this current manuscript correctly describes the status of NCL in *Drosophila* as "unclear". While reference #23 in line 107 (Lee et al. 2016) describes nucleolin in *Drosophila*, these authors expressed human nucleolin in *Drosophila* cells, and the NCL orthologue they cross-referenced in their Figure 2 is Cysteine string protein (Csp) which encodes a synaptic vesicle-associated co-chaperone of Hsc70 that is vital for presynaptic proteostasis and maintenance of synaptic function. In other words, Csp has nothing to do with nucleoli. I have no idea what the authors of reference 23 are talking about.

We thank the reviewer for catching this conflict and agree that this sentence is best left out. The reference has also been removed.

3) Deleting NUCL-1 did not cause lethality, but deletion caused delayed development. Do you think another GC protein (e.g., LPD-6) could partially substitute for NUCL-1 in function?

Thank you for this question. We do agree that NUCL-1 may share redundancy with other nucleolar RG/RGG proteins, particularly those that localize to the GC. With this possibility in mind, we have crossed the Δ NUCL-1 strain with the LPD-6 Δ RGG strain and assayed fertility (Figure 7J). While Δ NUCL-1 and LPD-6 Δ RGG worms show no defect in brood size compared to WT, Δ NUCL-1;LPD-6 Δ RGG worms have decreased brood size. This result indicates LPD-6 may partially compensate for NUCL-1 function.

4) Deleting the RGG/RG domain of NUCL-1 did not affect NUCL-1 Δ RGG's localization to nucleoli, but it did disrupt nucleolar morphology, causing Fibrillarin to redistribute partially from the FCs. In Line 158, you say loss of NUCL's RGG/RG domain "may not impact nucleolar function." This was based on the observation that there was no apparent change in nucleolar volume. But you still haven't tested nucleolar function in the absence of NUCL-1's RGG/RG domain. I would suggest deleting the phrase in lines 157-159 ("suggesting that loss...").

The reviewer is correct in pointing out that we have not directly tested nucleolar function. We have added the phrase, "as a loose proxy for nucleolar function" to describe our analysis of nucleolar volume (lines 231-235).

5) Did GARR-1 redistribute from the FC's with the loss of the NUCL-1 RGG/RG domain?

This is a great question that we would like to answer. However, we did not cross GARR-1 strains with the NUCL-1 strains because both genes are on the same chromosome, making crosses difficult. In addition, both GARR-1 and NUCL-1 Δ RGG are tagged with GFP11, therefore we cannot visualize both proteins at once with our current system. Future plans include tagging GARR-1 with wrmScarlet in our NUCL-1 lines to test if GARR-1 redistributes from the DFC with loss of the NUCL-1 RG/RGG domain.

6) Just curious, have you looked for methylation (asymmetrical di-methylation) on these RGG/RG motifs in the C. elegans proteins?

We have not yet evaluated methylation on RG/RGG motifs, but it is a future plan. We are currently deciding how to best test the possibility that RG/RGG methylation contributes to sub-nucleolar organization (ex: prmt mutants, RNAi, the auxin/degron system).

7) The genetic interactions between NUCL-1 Δ RGG and FIB Δ RGG in Figure 3 suggest a partial redundancy of their respective RGG/RG domains. Are you saying the RGG/RG domain of NUCL-1 can functionally substitute for the RGG/RG domain in Fibrillarin even when the two proteins occupy different sub-regions of the nucleoli? The concern I have is: Fibrillarin is an integral component of C/D box snoRNP complexes, and its RGG/RG domain may be important for assembly and function of these snoRNPs, and not so much for overall nucleolar structure. Perhaps GARR1's two RGG/RG domains are responsible for FC structure instead of Fibrillarin's single RGG/RG domain.

We thank the reviewer for bringing up this point and have added the possibility that the FIB-1 and GARR-1 severity of phenotypes may be in part due to their RG/RGG domains functioning in snoRNP assembly (lines 384-387).

Have you looked at nucleolar structure when just FIB Δ RGG is expressed in an otherwise WT NUCL-1 background? Admittedly, interpreting results from this experiment may be difficult as a loss in C/D box snoRNPs may be catastrophic for nucleolar structure and function.

FIB-1 Δ RG/RGG worms are not viable as homozygotes, thus we cannot perform this experiment.

8) Do you think a critical threshold concentration of RGG/RG's is required for nucleolar assembly (LLPS), and it doesn't matter which protein (NUCL-1, GARR1, or LPD-6) provides the RGG/RG motifs for LLPS? This might explain why deleting the RGG's from either NUCL-1 or LPD-6 had no drastic effect because enough were still provided by the other proteins present.

Thank you for bringing up this point. We have now expanded our discussion section to include our thoughts on this (lines 391-402).

REVIEWERS' COMMENTS

Reviewer #1 (Remarks to the Author):

The authors have addressed the reviewers comments with reorganization of the manuscripts and new data. The manuscript is much improved and will be of general interest.

Our only final suggestion is to clarify the model figure to show that the RGG mutant still maintains nucleolar domains - just on a finer scale. Video S4 clearly shows the micro domains - these should be shown in the model figure.

Also the authors should refer to this recent editorial which also questions the functional significance of nucleolar organization with respect to ribosome biogenesis <http://genesdev.cshlp.org/content/36/13-14/765.abstract>

Reviewer #2 (Remarks to the Author):

This revision from the Updike lab includes significant changes that strengthen the original, well-written manuscript. The new areas of discussion combined with additional results in the form of new figures and videos, thoroughly address all of my comments and suggestions. To my eye, it appears they have addressed all reviewer concerns, with the exception of a few non-essential experiments that are either not possible or not straightforward to complete.

This fascinating study will of high interest to researchers in the fields of RNA biology, cell biology, and those studying the role of nucleoli in neurodegenerative disease and cancer.

Reviewer #3 (Remarks to the Author):

The revised manuscript has addressed all my concerns and is ready to be accepted for publication.

Reviewer #4 (Remarks to the Author):

I think the revision is fine. But you might take a look at Alan Tartakoff's recent (2022) Perspective in Genes & Development (36:765-769) where he describes nucleolar structure and function in yeast, but as a counter-point to Liquid-Liquid Phase Separation.